# An intercomparison of Large-Eddy Simulations of the Martian daytime convective boundary layer

Tanguy Bertrand<sup>1</sup>, Aymeric Spiga<sup>1</sup>, Scot Rafkin<sup>2</sup>, Arnaud Colaitis<sup>1</sup>, François Forget<sup>1</sup>, and Ehouarn Millour<sup>1</sup>

<sup>1</sup>Laboratoire de Météorologie Dynamique (UPMC/CNRS), Paris, France <sup>2</sup>SouthWest Research Institute, Boulder, CO, USA

Correspondence to: Tanguy Bertrand (tanguy.bertrand -a- lmd.jussieu.fr)

**Abstract.** Large-Eddy Simulations (LES) for Mars resolve the Planetary Boundary Layer (PBL) turbulent dynamics by using a very fine horizontal resolution of a few tens of meters. LES modeling is becoming a more and more useful tool to prepare the robotic exploration of Mars by providing means to evaluate the intensity of convective plumes and vortices, horizontal wind gustiness, and turbulent fluctuations of temperature in the Martian PBL. In such context, and given the relative paucity

- of turbulence-related measurements on Mars, an intercomparison of LES models is a fruitful way to evaluate the models' predictions and to indicate possible areas of improvement. Thus, to prepare the landing of the ExoMars Schiaparelli lander (also named ExoMars Demonstrator Module, EDM), scheduled for October 2016, the results of the Laboratoire de Météorologie Dynamique (LMD) and South-West Research Institute (SwRI) LES models have been compared. The objective of this study is to determine the range of uncertainties, and dispersions, of the two numerical models' predictions, for the critical phase of the
- spacecraft's descent in the Martian daytime turbulent PBL. First, a strategy is defined to ensure similar radiative forcing in both the LMD and SwRI models. Then, LES are performed over a flat terrain with and without large-scale ambient horizontal wind. The LMD and SwRI Martian LES models predict similar temporal evolution of the PBL and organization in the horizontal and vertical wind fields. However, the convective motions in the daytime PBL are more vigorous by a factor 1.5-2 in SwRI results than in LMD results, independently of the presence or not of ambient horizontal wind. This discrepancy is further investigated
- through sensitivity studies to surface conditions, ambient wind, and airborne dust loading.

# 1 Introduction

In the Martian atmosphere, the Planetary Boundary Layer (PBL) depth can reach about ten kilometres above the surface in the daytime, when it is prone to intense turbulent convection associated with strong radiative warming of the surface (e.g., Gierasch and Goody, 1968; Sutton et al., 1978; Schofield et al., 1997; Smith et al., 2006). Conversely, in the night, convective motions

are inhibited by surface radiative cooling, which creates a near-surface stable layer, subsequently removed within about an hour after sunrise. As a result, the depth of the daytime PBL on Mars undergoes strong variations with both incoming sunlight (as determined by local time, season, and dust opacity) and surface thermophysical properties (thermal inertia and albedo).

Furthermore, the regional variability of the PBL depth on Mars is controlled by altimetry (Hinson et al., 2008; Spiga et al., 2010), as well as ambient (i.e. regional and large-scale) wind (Tyler et al., 2008).

The turbulent PBL is a very dynamic part of the Martian atmosphere, characterized by abrupt changes in temperature, pressure, wind, and aerosols. These PBL variations have been characterized mostly by instrumented landers and rovers such as the Viking spacecraft in the late 1970s (Sutton et al., 1978), Pathfinder in the late 1990s (Schofield et al., 1997), Mars

- as the Viking spacecraft in the late 1970s (Sutton et al., 1978), Pathfinder in the late 1990s (Schofield et al., 1997), Mars Exploration Rovers and Mars Phoenix in the 2000s (Smith et al., 2006; Whiteway et al., 2009) while Martian orbiters provided measurements of PBL depth (Hinson et al., 2008), as well as imagery of PBL-induced phenomena such as dust devils (Thomas and Gierasch, 1985) and cloud streets (Malin and Edgett, 2001). Despite this set of observations, the coverage of the Martian PBL activity has remained sparse thus far compared to e.g. what has been made available for the Earth's PBL.
- A complementary approach to observations to study Mars' PBL is the use of numerical modeling. Three-dimensional turbulence-resolving Large-Eddy Simulations (LES) have been employed since the early 2000s to assess the intense day-time PBL dynamics on Mars (Rafkin et al., 2001; Toigo and Richardson, 2003; Spiga and Forget, 2009; Gheynani and Taylor, 2011), which includes convective vortices generating the observed dust devils (Spiga et al., 2016, for a review). In LES, using a horizontal resolution of some tens of meters permits to resolve the larger turbulent eddies, which are the source of most of the
- energy transport within the PBL (Lilly, 1962). Given the relative paucity of PBL measurements on Mars, LES models are of crucial importance to help understanding the PBL processes, thus broadening our knowledge of the atmospheric circulations on Mars at all scales (from planetary scales to turbulent scales). LES have also become powerful tools to prepare Martian exploration, since they are being used to characterize atmospheric hazards in the Entry, Descent and Landing (EDL) phase of a spacecraft, and in turn to help define the design of the landing system (Kass et al., 2003; Rafkin and Michaels, 2003; Tyler et al. 2008).

et al., 2008).

At the time of writing, several LES models for Mars are available in the community to characterize Martian PBL dynamics (cf. reference above, plus Michaels and Rafkin, 2004; Richardson et al., 2007; Tyler et al., 2008; Spiga et al., 2010). All published LES describe the same qualitative behaviour: a deeply convective PBL during daytime, starting with a gradual growth of the mixed layer in the morning and ending with a rapid stabilization in the end of the afternoon, associated with polygonal horizontal cells, thermal plumes and convective vortices. However, not only the various models do not share the same characteristics, but quantitative estimates were found to differ when predictions from various Martian LES models were compared for EDL studies (Kass et al., 2003). A systematic intercomparison between at least two existing Martian LES models is still yet to be carried out to further characterize those differences. This is what is proposed in the present study.

The need to evaluate the differences predicted by two distinct Martian LES is threefold:

Contrary to observational data, an estimate of the uncertainties of LES predictions is still lacking. This is especially critical for the studies of atmospheric hazards during EDL, given the central role that LES predictions play in those studies and the relative paucity of available data to characterize the Martian PBL dynamics (e.g. for vertical winds). Comparing LES models would thus enable to identify the uncertainties in their diagnostics, thereby enabling an optimal EDL design for both landing spacecraft and definition of landing ellipse.

- 2. Carrying out a LES intercomparison would highlight discrepancies between results and help to identify the specific areas in which model improvements would be the most helpful. This overarching goal is beneficial for the whole Martian science. For instance, turbulent wind variability (i.e. "gustiness") plays an important role in controlling dust lifting on Mars (e.g., Mulholland et al., 2015, and references therein). Since turbulent wind measurements on Mars are very incomplete, LES predictions are still being an important source to assess the wind conditions on Mars associated with dust lifting (Fenton and Michaels, 2010). More generally, the continued development of Martian LES models is also of crucial importance to better understand the mechanisms responsible for heat and momentum transfer both by daytime PBL mixing and surface-atmosphere interactions. Following the tendency drawn by terrestrial studies, Martian LES predictions are more and more used to build and improve PBL parameterizations in Global Climate Models (GCMs) for Mars (Colaïtis et al., 2013).
- 3. Since Martian LES rely on hydrodynamical solvers inherited from terrestrial studies, confronting those models to the intense PBL convection on Mars (compared to the Earth, cf. Spiga, 2011) provides a stringent test for those solvers. An intercomparison study of Martian LES will ultimately be a strong driver of improvement for the atmospheric models used to carry out LES on Earth, and in an increasingly diverse range of planetary conditions (e.g. Venus LES, Barth and Rafkin, 2007; Imamura et al., 2014; Lefèvre et al., 2016).

15

20

In this paper, we compare the LES results obtained by, on the one hand, the Laboratoire de Météorologie Dynamique (LMD) Martian mesoscale model (Spiga and Forget, 2009; Spiga et al., 2010) and, on the other hand, the SouthWest Research Institute (SwRI) Martian mesoscale model (Rafkin et al., 2001; Michaels and Rafkin, 2004). We develop a strategy which makes our intercomparison study the first one of its kind for Mars: we ensure that similar physical constants and radiative forcing are employed in both models before performing a comparative analysis of LES results and conclude on the performance of the two dynamical solvers in predicting Mars' PBL convective motions. We further complement this intercomparison study by an exploration of the sensitivity of the convective PBL predicted by the LMD LES to surface thermophysical properties (e.g. albedo), ambient wind, and atmospheric dust loading.

We perform the present LES intercomparison in the context of the European Space Agency (ESA) ExoMars 2016 mission (hereinafter referred as ExoMars) with the aim of providing constraints for the EDL of the ExoMars Demonstrator Module (EDM, also named Schiaparelli). LES modeling is therefore performed at the ExoMars landing site, namely in the Terra Meridiani region (latitude  $-1.82^{\circ}$ N, longitude  $-6.15^{\circ}$ E), for the landing scheduled in northern autumn (solar longitude  $L_s = 244^{\circ}$ ).

In Section 2, we describe the LMD and SwRI models used in this LES intercomparison. In section 3, we provide details on the intercomparison strategy, and how we reached similar radiative forcing both in SwRI and LMD LES. The results of both the Martian LES intercomparison and sensitivity study are discussed in section 4. We conclude about the similarities vs. discrepancies between the two models in section 6.

# 2 Models description

We provide here the key points to describe the two models used for our LES intercomparison. Further details about each model can be found in the references provided in this section.

Both LMD and SwRI Martian LES models have been built independently by adapting terrestrial mesoscale models to the 5 Martian case, with the coupling of specific physical models (namely, radiative transfer and soil model) initially developed for Martian GCMs:

- LMD LES are performed using the LMD Martian Mesoscale Model (Spiga and Forget, 2009; Spiga et al., 2010), based on the Weather Research and Forecast (WRF) model and its fully compressible non-hydrostatic dynamical core (Moeng et al., 2007; Skamarock and Klemp, 2008), combined with the comprehensive set of physical parametrizations of the LMD GCM (Forget et al., 1999).
- SwRI LES are performed using the Mars Regional Atmospheric Modeling System (MRAMS), a nonhydrostatic Martian mesoscale model developed at SwRI (Rafkin et al., 2001; Michaels and Rafkin, 2004) and based on the terrestrial RAMS dynamical core (Pielke et al., 1992), which physical parameterizations are inherited from the Martian NASA Ames GCM (Haberle et al., 1993).
- The two LES models not only use very distinct different radiative transfer and soil model (inherited from GCM), but also use different dust scattering properties (Ockert-Bell et al. (1997) for SwRI LES vs. Wolff et al. (2009) for LMD LES), which can lead to significant departures in the predictions of atmospheric temperatures.

Although the largest turbulent eddies (contrary to global and regional climate models) are resolved, LES still lack the mixing produced by the unresolved small-scale eddies, which requires the inclusion of parameterizations for subgrid-scale diffusion.

- In the two models used for this intercomparison, the resolved large-eddy Turbulent Kinetic Energy (TKE) can be used to assess the strength of small-scale mixing in the parameterization, which is usually effective on the three spatial coordinates rather than the sole vertical dimension. The SwRI Martian LES model contains a specific Deardorff diffusion scheme (Deardorff, 1980), also used in the terrestrial version of MRAMS. The LMD LES model uses the strategy adopted for WRF terrestrial LES (Moeng et al., 2007), which is similar qualitatively to the one adopted by SwRI with MRAMS – although subgrid-scale mixing
- coefficients differ between LMD and SwRI LES (see Section 4 and 6). Both models use a (qualitatively similar) Richardsonbased surface layer to compute surface-atmosphere transfers of heat, momentum, and tracers (transfer coefficients vary with atmospheric stability, see e.g. Colaïtis et al., 2013).

Although the dynamical and physical parts of the LMD and SwRI LES models are different, the simulation framework in both models is similar. To resolve the 3-D convective plumes, cells and vortices, LES are performed on a domain using

periodic boundary conditions, with horizontal and vertical resolutions of a few tens of meters. The radiative transfer models in LES are combined with a horizontally uniform and static dust profile. Surface properties (topography, albedo, thermal inertia) are uniform too in the LES domain. The models typically compute 10 Martian hours during daytime to capture the convective

PBL rise, growth and collapse. In both the LMD and SwRI models, a random (noise) perturbation of 0.1 K amplitude is added to the initial temperature field to break its symmetry and help trigger convective motions (Michaels and Rafkin, 2004).

# 3 Intercomparison methodology

Since the LMD and SwRI models use a combination of very distinct radiative transfer modeling, dust properties, and subsurface modeling, a preliminary preparation of models is necessary to prevent the LES intercomparison from being simply an intercomparison of radiative transfer schemes, in the radiatively-controlled (by dust and CO<sub>2</sub>) Martian environment. This is actually one of the reasons why a systematic intercomparison study between mesoscale models or LES for Mars has not been carried out yet: thus far the models used for comparisons did not use similar radiative forcing, making the analysis of dynamical differences rather cumbersome, although the dispersion of simulated results remained informative about differences between

- models (e.g. Kass et al., 2003). Conversely, what is aimed in the present intercomparison study is to assess the departures in PBL dynamics possibly arising from the use of two distinct LES models rather than departures resulting from combined differences in the dynamical core and physical parameterizations. To that end, not only we use the closest possible LES settings between LMD and SwRI modeling framework (section 3.1), but we also ensure that the same radiative forcing of the atmospheric flow is imposed in the LMD and SwRI models (section 3.2), thereby enabling us to conduct consistent dynamical
- comparisons of both LES models in section 4.

#### 3.1 General settings

The main settings used in LMD and SwRI LES are summarized in Tables 1 (model parameters) and 2 (planetary constants). The rationale for those choices is to reach a high level of similarity in both models as far as model domain, planetary constants and initial conditions are concerned.

- The computational domain has to be wide enough to contain several convective cells (Mason, 1989), in order to derive consistent PBL statistics from LES results. At the same time, the horizontal resolution must be fine enough to enable a good representation of the "large eddy" part of the turbulence spectra. Hence the choice for an horizontal resolution of 50 m and 145× 145 grid points (this configuration provides similar PBL statistics as other configurations with more grid points, as is detailed in Spiga et al., 2010). A model top at 12 km above the local surface is used, high enough to ensure that the boundary layer 25 growth will not be influenced by the position of this model top.
- The LES models are run with 150 (SwBI) and 201 (LMD) vertical lavels

The LES models are run with 150 (SwRI) and 201 (LMD) vertical levels, which yields a vertical resolution of approximately 80 m and 60 m respectively in the vertical dimension. Given the type of vertical coordinates employed in each model (WRF uses mass-based coordinates, a slightly different version of sigma coordinates than what is used in RAMS), this difference of vertical resolution is imposed by the typical set of vertical levels which optimizes the performance of physical parameterizations

for each model. However, based on the sensitivity study carried out in Spiga et al. (2010), this difference in vertical resolution does not affect significantly the PBL predictions through LES. The key requirement is that the vertical resolution is refined to a few meters close to the surface, which is ensured in both models.

| Parameter                   | Model settings for the intercomparison                                                          |
|-----------------------------|-------------------------------------------------------------------------------------------------|
| Horizontal grid (x,y)       | $145 \times 145$                                                                                |
| Horizontal resolution       | 50 m                                                                                            |
| Vertical grid               | 201 levels (LMD) - 150 levels (SwRI)                                                            |
| Vertical resolution         | 60 m (LMD) - 80 m (SwRI)                                                                        |
| Model top                   | 12 km                                                                                           |
| Dynamical time step         | 0.5 s                                                                                           |
| Region                      | -1.82°N; -6.16°E (Meridiani Planum - ExoMars 2016 landing site)                                 |
| Solar Longitude             | 244° (Landing date, northern autumn)                                                            |
| Surface conditions          | Albedo: 0.21 - Thermal inertia: 238 tiu (adjusted to 300 tiu in the LMD LES, see Section 3.2)   |
| Dust vertical distribution  | Conrath type: $P_0 = 610$ Pa and $\nu = 0.007$                                                  |
| Dust opacity                | $\tau = 0.2$ (horizontally uniform and constant over time)                                      |
| Initial temperature profile | Extracted from LMD-GCM run with uniform dust loading of $\tau$ =0.2 at geographical coordinates |
| initial temperature prome   | and $L_s$ of the ExoMars reference landing site (Meridiani region)                              |
| Ambient wind                | Two cases: 0 and 15 m s <sup><math>-1</math></sup>                                              |

Table 1. LMD and SwRI LES models settings and configurations for the intercomparison.

| $c_p (\mathrm{J.kg}^{-1}.\mathrm{K}^{-1})$ | $r (J.kg^{-1}.K^{-1})$ | $g (\mathrm{m.s}^{-2})$ | $\epsilon$     | $s_{hc}$ (J.kg <sup>-1</sup> .K <sup>-1</sup> ) | $s_{rho}$ (kg.m <sup>-3</sup> ) | $D_s$ (s) |
|--------------------------------------------|------------------------|-------------------------|----------------|-------------------------------------------------|---------------------------------|-----------|
| 770                                        | 192                    | 3.72                    | 0.99           | 711                                             | 1500                            | 88775.244 |
| $\Phi$                                     | e                      | sma (AU)                | $L_{yr}$ (sol) | $S_0 (W.m^{-2})$                                | $\Omega (\mathrm{rad.s}^{-1})$  | $R_M$ (m) |
| 25.1919                                    | 0.09341233             | 1.52366231              | 669            | 1367                                            | 7.08821e-5                      | 3390000   |

**Table 2.** Physical and planetary constants for the Martian atmosphere, as defined in the models:  $c_p$  is the specific heat capacity, r is the specific gas constant, g is the gravitational acceleration,  $\epsilon$  is the ground emissivity,  $s_{hc}$  is the subsurface specific heat and  $s_{rho}$  is the subsurface density.  $D_s$  is the duration of a sol,  $\Phi$  is the planet obliquity, e is the eccentricity of the planet orbit, sma is its semi major axis,  $L_{yr}$  is the number of sols in one Martian year,  $S_0$  is the solar constant at 1 AU,  $\Omega$  is the planetary rotation rate and  $R_M$  is Mars radius.

For the chosen grid spacing, the dynamical time step is set to be small enough to ensure numerical stability according to CFL criterion, but it has also to be high enough to reduce the usually expensive computational cost of LES – especially given that sensitivity runs are included in this intercomparison study. Thus, a good trade-off for the LES timestep has been found to be 0.5 s. LES are carried out from local times 06:00 to 18:00. The comparison of the results is mainly performed between

5

11:00 to 17:00 local time since convective motions are usually amongst the strongest around these local times. Furthermore, this range was convenient to assess ExoMars landing atmospheric conditions since the spacecraft is designed to land in the local-time window of 14:00-16:00.

As is mentioned in the introduction, the LES were performed to assess atmospheric hazards for the ExoMars mission. Therefore both surface thermophysical properties and initial temperature profile reflect the conditions at the expected season (end of

northern autumn, areocentric longitude  $L_s = 244^{\circ}$ ) and location (Terra Meridiani, latitude  $-1.82^{\circ}$ N, longitude  $-6.15^{\circ}$ E) for the ExoMars landing. Thermal inertia is set to 238 J s<sup> $-\frac{1}{2}$ </sup> m<sup>-2</sup> K<sup>-1</sup> (unit hereinafter referred to as tiu for "thermal inertia unit") and albedo to 0.21 – those values are extracted from the Thermal Emission Spectrometer (TES) nighttime data, averaged over a 1000 km<sup>2</sup> location around the ExoMars landing site. LMD and SwRI runs use the same initial temperature profile at 06 : 00

- local time, extracted at the season and location of ExoMars' landing, from a run with the LMD GCM (Forget et al., 1999) performed with a constant and uniform dust opacity of 0.2, as well as a Conrath-type dust vertical distribution (as is detailed in Pollack et al., 1990; Forget et al., 1999). Both LMD and SwRI LES also use these settings for dust opacity and dust vertical distribution. Setting a dust opacity of 0.2 might be an underestimate for the actual value at this season on Mars (Montabone et al., 2015), which ranges 0.3 0.5, but assuming a clear atmosphere ensures the most unstable situation, thus the maximum
- strength for convection in LES runs (see Figure 17). Since this study's focus is on highlighting differences in convective PBL predictions, a more convective situation yields a more stringent test for LES models.

# 3.2 Reaching a similar radiative forcing in the two compared models

The two LES models we aim to compare share a fair amount of similar settings, as detailed in section 3.1. As is discussed in the beginning of this section 3, this similarity is necessary but not sufficient to ensure a consistent intercomparison of the

- LES predictions for the PBL dynamics on Mars. Despite our efforts, the fact that the LMD and SwRI models employ different radiative transfer schemes and dust properties leads to large temperature differences between the models, which overwhelms – or at least competes with – dynamical differences. It would thus be challenging to ascribe the intercomparison differences to either radiative or dynamical differences.
- A complete rewrite of either the LMD or SwRI LES to couple the same set of physical parameterizations is not possible in a 20 reasonable amount of time. Hence the solution we found is to modify a small amount of key parameters in the LMD radiative scheme, so that it replicates SwRI radiative transfer predictions and ensures that the radiative forcing of the PBL in the LMD and SwRI LES are similar. Those key parameters are mostly related to dust optical properties (given the strong radiative control of dust on the Martian atmospheric structure, e.g. Gierasch and Goody (1972)) and surface thermophysical properties (albedo, thermal inertia), which controls the surface thermal balance in the Martian environment where atmospheric density is so low 25 that the impact of sensible heat flux on this balance is not prominent (e.g. Savijärvi and Kauhanen, 2008).

We performed the comparisons and corrections using the single-column version (1D) of the respective physical parameterizations used in the LMD and SwRI LES models. All settings of those 1D models are similar to what is described in Tables 1 and 2 for the LES runs, and correspond to ExoMars landing location and season. For the sake of comparison, the same fine vertical resolution has been chosen for simulations with the LMD and SwRI models. The 1D models do not integrate any

30 hydrodynamical equation: an ambient wind of  $10 \text{ m s}^{-1}$  is only imposed to obtain a realistic value for the sensible heat flux, whose influence on surface temperature is small but not negligible. Convective adjustment, that is usually included in 1D models to model mixing by turbulent plumes resolved by LES, is turned off in the 1D models as is the case in LES integrations. The radiative response of the LMD and SwRI 1D models is assessed by comparing the obtained equilibrium temperature profiles.

Figure 1. Temperature profiles obtained by LMD (red) before any adjustment and SwRI (black) 1D models at the ExoMars landing site and season. Left are temperatures at local time 00:00, right are temperatures at 12:00.

A match between the two profiles predicted by the 1D models means that the PBL dynamics in both the LMD and SwRI LES are forced by similar (unstable) gradients of vertical temperature imposed by radiative forcing.

The same initial temperature profile is used in the LMD and SwRI 1D models. This initial profile was obtained by running the LMD 1D model during 50 Martian days (repeating the same day corresponding to  $L_s = 244^{\circ}$  conditions). A simulation duration of 50 Martian days is large enough to reach steady-state equilibrium given the short radiative timescales of the Martian atmosphere (this was checked in practice in our 1D simulations). This profile is then used to initialize both the LMD and SwRI models, which are subsequently run for an additional 50 Martian days to ensure that the steady-state equilibrium is reached. In any case, although using the same initial temperature profile is helpful for consistency, results are not very sensitive to the

In Figure 1 and Figure 2, we show the two temperature profiles and the surface temperatures obtained by the LMD and SwRI 1D models prior to any correction. To first order, the temperature profiles appear similar. Nonetheless, atmospheric temperatures in the LMD model are found to be 5 - 10 K colder than SwRI temperatures. This bias extends over almost the entire atmospheric column. In addition, LMD daytime surface temperatures are up to 8 K warmer than SwRI results. This further justifies the approach adopted in this section 3.2.

assumed initial profile, owing to (again) the short radiative timescales of the Martian atmosphere.

- Our working assumption is that, given the central role played by dust on controlling the Martian thermal structure, modifying key dust radiative properties would enable us to obtain a match between the temperature profiles predicted by the LMD and SwRI models. We chose to vary 1. the dust extinction efficiency  $Q_{ext}$  (or "thermal infrared opacity", at the reference infrared wavelength) and 2. the visible single scattering albedo  $w_0$ , which quantifies dust "brightness" (further details and explanations in Forget et al., 1999; Madeleine et al., 2011). For the sake of simplicity, we define the ratios S and C of the corrected value
- 20 over the reference value for respectively  $w_0$  and  $Q_{ext}$  in the LMD model. The higher S, the brighter dust; the higher C, the stronger absorption by dust in the infrared.