# Peer review of "An intercomparison of Large-Eddy Simulations of the Martian daytime convective boundary layer"

_Geoscientific Model Development, 2016_

## Referee Comment (RC1) · M. Mischna (Referee) · 15 Dec 2016

The manuscript, "An intercomparison of large-eddy simulations of the martian daytime convective boundary layer" by Bertrand, Spiga, Rafkin, Colaitis, Forget and Millour aims to compare results from two independent large eddy simulation (LES) models, one operated by the Laboratoire Meteorologie Dynamique (LMD) and one by the Southwest Research Institute (SwRI). The intent is to identify areas in which LES models can be improved, based on differences in their output for similar conditions. Ultimately, this will yield a better product for entry, descent and landing (EDL) activities of future spacecraft. I think that this manuscript can positively advance our understanding of LES modeling, and its application to EDL, however in its present form, it suffers from

too many shortfalls to be acceptable without some substantive changes. The authors are on the right track, but disappointingly leave many areas of fruitful investigation unaddressed and incomplete. None of them are fatal to the manuscript, but they are not trivial to address, either. I believe if these particular issues are properly addressed, the manuscript will be greatly strengthened, and will serve as a go-to reference for the EDL modeling community.

High-level comments:

Pg. 2, line 21: You identify more than two extant LES models, the present ones, plus the MarsWRF LES and the OSU LES. As you note, the need to evaluate the differences predicted by distinct martian LES are many-fold. It would seem to me, then, that doing an intercomparison with only half of the available LES models makes this study somewhat incomplete. In particular, the LMD LES is based on the WRF framework, as is the MarsWRF LES, although they have been developed independently. That seems ripe for comparison. Were these other two groups approached to contribute to the intercomparison and, if not, why not? This, to me, is a significant weakness of the manuscript. It's less an intercomparison of Mars LES, and more a simple comparison of two Mars LES.

Section 5: I'm somewhat uncomfortable with this section in that you only perform sensitivity studies on one of the two models (the LMD model). Âă This, then, becomes less of a model intercomparison and more of a sensitivity study of a single model. Âă The two are quite different, and I would argue that the intercomparison study essentially ends in Section 4. Âă I would like to see similar sensitivity studies for the SwRI model to evaluate whether, for example, the greater resolved TKE, or vertical wind speeds are more sensitive to parameter changes in the SwRI model than in the LMD model.

Specific comments:

Pg. 5, line 24: Can you expand on the validity of this assumption? In mesoscale modeling, there are 10s of km between the top of the 'good' results from the model,

and the domain top. It seems that you're getting pretty close to the top of the domain when looking at the PBL, which comes in only a couple km below the model top. Are there issues with damping layers at the model top that might be affecting your results?

Figure 4: While you argue that the radiative forcing is about the same now after doing the radiative adjustment, the near-surface atmospheric temperatures are still vastly different–10 K in the nighttime and >20 K in the daytime based on this figure.Ăă I can only imagine that is going to have a noticeable effect on the magnitude of turbulent activity at the smallest scales nearest to the surface.Ăă I don't see any discussion or acknowledgement of this difference. Surely it has to be important!

Pg. 11, line 28: You discuss 'quantitative discrepancies' between the models as being responsible for some of the differences between LMD and SwRI, and then refer forward to Section 6. I think it needs to be stated here what these quantitative differences are for the reader to understand and interpret the results of this section.

Pg. 14, line 2: To be honest, I don't think you've done any investigation of the discrepancies between the two models to this point. You've identified what they are, but you haven't done any interpretation of what is causing those discrepancies, or provided any insight into how they might be resolved.

Pg. 16, line 21: This in an incomplete comparison–what were the model parameters in this 'other' LMD LES simulation?Ăă It's peculiar to say that the current results are comparable to past results that you don't show, because I, as a reader, have no objective way to assess that statement.Ăă What defines 'good agreement', for example?

Pg. 16, line 28: This is tied into the 'quantitative discrepancies' comment on Pg. 11, above. The difference in subgrid-scale diffusion scheme seems like a key difference that has gone unexplored.Ăă You've already gone to the effort to match as many physical parameters of the two models as you can, so why not the subgrid-scale diffusion scheme?Ăă It's somewhat of a cop-out to say that you see differences between the models, and then speculate on what might cause those differences (subgrid-scale dif-

[Figure]

fusion) without trying to actually determine if it is, indeed, a cause. I think this study is incomplete because of this.

Figure 5: Can you explain why there are far more points in the LMD curves than in the SwRI curves if both models have the same timestep?Âă Is it as simple as more frequent output in LMD vs. SwRI?Âă If so, why was that not coordinated?Âă The higher frequency output of LMD gives the impression it is 'noisier' than SwRI, and it should probably be reduced to the same output frequency for plotting, if this is a rigorous intercomparison.

Figures 10, 11, 12, 13, 16: Can you explain why in all of these figures, the SwRI LES output is truncated before the end of the time period under investigation? Also, in Figure 10, the LMD data is truncated at 17:00 as well. These figures need to be complete, or else an explanation given for their incompleteness. If it is due to something like a model crash, then this needs to be investigated and explained. I would not feel confident at all in model results that derived from a simulation that crashed. If it's just because the model was stopped because the interesting results had finished at a particular time, then this also needs to be explained and/or made consistent across all panels.

Typographical/minor issues:

Pg. 2, line 14: Change 'to resolve' to 'resolution of'

Pg. 2, line 25: Insert 'do' between 'only' and 'the'

Pg. 3, line 17: Change 'SouthWest' to 'Southwest'

Pg. 3, line 29: Change 'Section 2' to 'section 2' to remain consistent with other sections

Pg. 3, line 32: You jump from describing section 4 to section 6 without mentioning section 5

Pg. 4, line 13: Insert 'in' before 'which'

Pg. 4, line 15: Change 'soil model' to 'soil models'

Pg. 4, line 20: Change 'Turbulent Kinetic Energy' to 'turbulent kinetic energy'

Pg. 5, line 10: Insert 'at' between 'aimed' and 'in'

Pg. 5, line 10: Change 'to assess' to 'an assessment of'

Pg. 9, line 1: Insert 'the' after 'above'

Pg. 9, line 2: Change '1D PBL' to '1D, the PBL'

Pg. 9, line 17: You mention Ames here, but the reader has no context as to what that means.

Pg. 9, line 19: Change 'of about' to 'by about'

Figure 3: You reference 'top' and 'bottom' but in this layout, it should be 'left' and 'right'

Pg. 11, line 9: Here, you've jumped from Figure 5 to Figure 8, and then to Figure 9, then back to Figures 6 and 7. Figures should go in the order they are referenced. (So Figure 8 should become Figure 6, Figure 9 should become Figure 7, and Figures 6 and 7 should become the new Figures 8 and 9.

Pg. 11, line 17: Insert 'a' between 'are' and 'typical'

Pg. 11, line 20: Change 'predict maximum updrafts values' to 'predicts a maximum updraft value'

Figure 8: Is this for the models with no background wind or 15 m/s background wind?

Pg. 12, line 14: Change 'to compare' to 'comparison of'

Pg. 12, line 21: Change 'radio-occultations, which are' to 'radio occultations, which is'

Pg. 12, line 28: Change 'condition' to 'conditions'

Pg. 14, line 1: Change 'This enables to better understand' to 'This enables better

understanding of'

Pg. 14, line 7: Change 'mixing coefficient' to 'mixing coefficients'

Pg. 14, line 9: Change 'Same conclusions' to 'The same conclusions'

Pg. 15, line 25: Change 'cause' to 'causes'

Pg. 16, line 16: Change 'or' to 'and'

Pg. 16, line 32: Insert 'the' between 'at' and 'grid'

Pg. 17, line 10: Change 'is' to 'are'

Figure 10 caption: Change '09:00' to '08:00'

Figure 5 goes from 09:00-17:00. Figure 10 goes from 08:00-17:00. Figure 11 goes from 07:00-19:00. Figures 12 and 13 go from 11:00-17:00. Figures 16-21 go from 08:00-17:00. Why not plot everything on the same temporal axis? Consistency makes the reader happy, and the manuscript easier to follow!

––––––––––––––––––––––––

---

## Referee Comment (RC2) · D. Tyler Jr. (Referee) · 25 Jan 2017

Review of:

Journal: GMD Title: An intercomparison of Large-Eddy Simulations of the Martian daytime convective boundary layer Author(s): Tanguy Bertrand et al. MS No.: gmd-2016-241 MS Type: Methods for assessment of models

reviewer: Daniel Tyler Jr. (College of Earth Ocean and Atmospheric Sciences; Oregon State University; Corvallis, OR, USA)

manuscript overview: This manuscript endeavors to accomplish two goals: 1) perform an intercomparison of the dynamical cores of the LMD and SwRI LES models, and 2)

provide a high-resolution LES assessment of the turbulent convective boundary layer for the ExoMars Schiaparelli lander during its afternoon EDL. The first of these goals is the primary one; the second was never directly addressed since no mesoscale forecasts for the landing site were considered/compared with the LES results.

review overview: I found that a good effort was put into this work by the authors; I'm encouraged to see a manuscript with a model intercomparison focus (this speaks to the awareness of what I believe is an important issue in our community). However, with only one simulation from the SwRI LES model to examine, accomplishing the stated goals is unlikely no matter the effort involved. Moreover, there is one overarching problem with this study: the grid used is entirely inappropriate in consideration of typical afternoon mixed layers on Mars (as based on the body of literature regarding LES). Thus, I must recommend rejection, although I enthusiastically recommend the authors redo the modeling on an appropriate grid, and the resubmit the manuscript upon consideration of my comments in 2) and 3) below.

ISSUES IDENTIFIED IN THIS REVIEW:

1) the horizontal size of the modeling domain used in this study (a fatal problem):

When performing LES studies of the convective boundary layer (CBL), it is imperative that the horizontal size of the domain is large enough so the periodic boundary conditions cannot influence/contaminate the solution computed for the domain interior. The authors suggest that they have followed the guidance of Mason (1989) to achieve this, but they haven't. For typical square LES domains, a generalized "rule of thumb" (as was used, if not clearly stated, by Mason (1989)) is to design the grid so the length of the domain side is ∼3x the size of the largest eddy that will be resolved by the simulation. For an afternoon Mars EDL related investigation, the size of the largest eddy scales as the maximum depth of the CBL, which can reach ∼10 km (thus, ∼30 km would be an appropriate lateral size). It's an unfortunate reality of LES for Mars that, to sufficiently resolve the range of smaller eddies and the complexity of convective structures (with the underlying desire being to capture the energy spectrum), a grid-spacing of ∼50 m (as was used here) is needed. For a square LES domain with a 50 m grid-spacing, the number of computational locations would then be 600x600. In this work, with only 145x145 computational locations, the lateral size of the domain is just 7.25 km (this is far too small), and the problems that will be created in this approach will contaminate the analysis and intercomparisons that could be performed at a time of day with a deep CBL. For Mars, especially once any wind profile is introduced to force the simulation, this ∼3x aspect ratio rule likely should be considered a minimum, and this is due to dramatic variations seen across the diurnal cycle of the scale of convective structures as a function of time and height in the domain. this poses a real problem for LES of the Martian atmosphere, and any study desiring a high-quality LES model intercomparison should err on the side of having completely eliminated the possibility of a contaminated simulation due to a very deep CBL that will interact with the periodic boundary conditions. Most certainly, the top of the modeling domain should also be well above the top of the CBL. As used here, a top of 12 km is probably sufficient, although a few km higher would be preferred. For the goals of this manuscript, the 145x145 number of computational locations, with a grid spacing of 50 m, is a fatal problem. It's easy to see that LES studies involving the CBL on the planet Mars almost certainly require the use of massively parallel architectures.

2) dust used during the intercomparison phase (thoughts/suggestion):

As described by the authors, without managing the issue of dust and its differing treatments between the two models, an intercomparison of LES models could reduce itself to an 'intercomparison of radiation schemes', not of the dynamical cores as is desired (this is also true for mesoscale and global climate model intercomparisons). The authors did put effort into getting the LMD model to show an improved agreement with the SwRI model to facilitate their intercomparison, although I wasn't fully convinced this effort was actually successful. From a perspective of do it as simply as possible, it's unclear why the authors didn't just run/compare the two models (the primary focus of this

Interactive
comment

manuscript) with no dust loading at all, a dust-free atmosphere. Even with dust properties modified so ground temperatures come into far better agreement, there was no discussion about heating rate profiles, that they had also come into better agreement as a result. It is much easier (and more straightforward) to use a dust-free atmosphere for the intercomparison focus of this effort, eliminating the complication of the radiative properties of dust in the atmosphere, and the probable non-linear response to this change the the heating rates as a function of height. For the secondary aspect of this effort (the prediction of the EDL environment for the Sciaparelli spacecraft), dust would be reintroduced in both models, and LES results would need to be compared with mesoscale model results (presumably form both of the parent models). Most certainly, a primary reason to use LES is to both improve and qualify our confidence in (and understanding of) the results from mesoscale models, specifically the performance of PBL schemes being used (since there is no PBL scheme in LES). Moreover, and I believe this is important, the use of mesoscale model results would allow the ability to characterize the larger-scale environment in which the LES was being performed. It's important because the larger-scale regional/local circulation can dramatically affect the evolution of the CBL. Careful consideration of this for site-specific LES is not addressed by any authors to date, whereas local strong subsidence has been shown by Tyler and Barnes (2015) to be very important to the development of the CBL, with actual evidence of this (Moores et al., 2015). How can this reality be incorporated into LES, and is it even possible? For Mars, LES has unique challenges, and a manuscript sufficiently well thought-out, that addresses some of these issues head-on with some new approaches would be most-welcomed and likely important to the community.

3) subgrid mixing parameterizations (thoughts/suggestion):

Upon looking at the results provided, it's easy to agree with the authors that subgrid mixing in the LMD model is much stronger than it is in the SwRI model. A short section in the manuscript suggested the authors did experiment with the subgrid mixing strength in the LMD model. Unfortunately no results were shown to indicate the degree
of change seen towards what I would expect to be a much more 'noisy' solution (more like that in the SwRI model). Were such changes seen in that exercise, and if not did the authors try to completely disable the subgrid mixing to insure it had indeed been modified? Analogously, subgrid mixing schemes are to LES as PBL schemes are to mesoscale models; and, both are fundamentally untested in regard to being used in atmospheric modeling for Mars (designed for and tested in terrestrial modeling). I believe we should have a bit more trust in the subgrid mixing scheme of LES (as being fundamental) than the PBL scheme of a mesoscale model, although doubt must still be raised in regard to the isotropic nature of vertical mixing compared to horizontal in a deep and energetic CBL. How do individual schemes address this? The results here serve to elucidate the central importance of subgrid mixing schemes in LES model intercomparisons, where a thorough intercomparison would investigate a wide range of strengths of subgrid mixing. We should be able to show that results become 'smooth' on one hand (as in the LMD results) and very 'noisy' on the other hand (as in the SwRI results) with variations in the strength of this mixing. The idea would be to investigate the point at which the energy spectrum became sufficiently corrupted and/or modified. Possibly, a balancing of the subgrid mixing schemes between the two models, in a no-dust scenario such that the energy spectra became more similar, would be a logical framework from which to begin this intercomparison. This has never even been tried by investigators, and as so clear in the results presented, it's a central question that was not sufficiently addressed.

MINOR THOUGHTS/ISSUES:

The vertical grid of the SwRI model is actually quite different, with heights fixed above ground in a sigma z (not sigma p) formulation, quite different form the vertical grid used in the LMD model. It actually leads to a "pressure cooker" effect in MRAMS. For comparison of these two models, it is worth acknowledging that there shouldn't be any surface pressure variation in the LMD model, while there should be in the SwRI model.

Without a doubt, spacecraft payloads are only going to get more massive, which renders the sensitivity of any spacecraft to atmospheric turbulence quantified during the EDL phase by LES in the CBL (such as for Phoenix or Insight) a moot point. The LES modeling that was done for Phoenix was driven by the possibility of oscillations in the parachute/lander system that (according to my understanding) are not problematic for either the MSL or the Mars 2020 spacecraft. I'm unaware of any LES modeling being requested/performed for Insight. In the near future, SpaceX and others will be landing massive payloads; and, the EDL trajectories for such payloads will cause the spatial variation of the CBL depth (as modified by topographic circulations) to be far more important during the final phase of 'flight' (quite low for large horizontal distances to make the greatest use of larger air densities). Results from LES are likely to provide the turbulence spectra that turns out to be crucial for autonomous navigation software, etc... My sense at this point is that the reason for high-resolution LES modeling may have much more to do with autonomous robotic flying exploration vehicles, such as Mars Helicopter: http://www.jpl.nasa.gov/news/news.php?feature=4457

The manuscript was not clear on whether the surface layer scheme truly only acts on the lowest model layer acting with the ground for turbulence closure. I know this is the case for the OSU Mars LES model and believe it must also be so for the LMD and SwRI LES model. It's a point worth making, especially since the authors see differences in the near surface wind and/or temperature profiles. The structure near the surface is quite possibly another means through which the subgrid mixing can be examined in an intercomparison.

The authors certainly recognize that systematic intercomparisons between mesoscale and/or LES models have not been carried out, possibly because no funding entity seems to think that such intercomparisons are important. I acknowledge and appreciate that this work has tried to remedy this situation. Since atmospheric dynamics are actually so much more important on Mars in relation to physical processes than they are on the Earth, raising awareness here is important. Possibly, we are stuck in a terrestrial modeling paradigm that we need to escape to improve our mesoscale and

LES modeling of the Martian atmosphere.

Finally, even though it only leads to greater computational burden, it is definitely worth considering the value of running the LES model through one full diurnal cycle before examining results from the second full diurnal cycle. Such 'spin-up' does affect the results, with the second diurnal period quite different. When background forcing is added, the second diurnal cycle can be quite different from the first, where the morning air temperature profiles (before the onset of convection) differ significantly, which noticeably affects the growth/evolution of the CBL. In the diurnal cycle, there is another weakness of LES (the highly stable regime, even more stable in the atmosphere of Mars) to investigate and better understand. We strive for the most realistic/honest results from our models; and, in this, intercomparison is an art form that I hope efforts such as this convince us to practice more enthusiastically.

Dan Tyler

―――――――――――――――――

---

## Author Comment (AC1) · 31 Mar 2017

Dear Mike,

Thank you for your careful and thorough reviewing of our manuscript and your insightful comments. We believe that those comments complements very well our manuscript by helping to identify the many challenges of model intercomparisons, an overarching goal truly difficult to address. Probably our submitted version reads as the definitive work on this topic, while this was absolutely not our intent. The goal of our paper is to provide the community with the report of a first attempt to thoroughly compare two LES models for Martian Planetary Boundary Layer convection. We corrected the paper accordingly to clearly reflect this modest goal and state the remaining goals.

We changed the title of the paper. We also added an entire section "Challenges of LES intercomparisons and suggestions for future studies" where we summarize the difficulties and ideas mentioned in the paper and in the reviews. We strongly believe the revised version will be a useful milestone for the community of Martian modelers, if not a definitive reference on Martian model intercomparisons, which is clearly an ambitious goal for a future study funded much more extensively than the preliminary one we performed.

We add attached as a comment a pdf document comparing the previous and the new version of the manuscript (in order to better track the changes). Note that the references are not displayed in this pdf document (made with latexdiff which does not take references into account and also get confused sometimes with the order of the sections) but of course, in the submitted paper, references and section ordering are fine.

**High-level comments: Pg. 2, line 21: You identify more than two extant LES models, the present ones, plus the MarsWRF LES and the OSU LES. As you note, the need to evaluate the differences predicted by distinct martian LES are many-fold. It would seem to me, then, that doing an intercomparison with only half of the available LES models makes this study somewhat incomplete. In particular, the LMD LES is based on the WRF framework, as is the MarsWRF LES, although they have been developed independently. That seems ripe for comparison. Were these other two groups approached to contribute to the intercomparison and, if not, why not? This, to me, is a significant weakness of the manuscript. It's less an intercomparison of Mars LES, and more a simple comparison of two Mars LES.**

We completely agree with this comment. Ideally, we would have compared all existing Martian LES at different seasons and local times. This would clearly be the best way to examine those models and would form a true intercomparison study. This would require far more human, time and funding resources than the one we were able to

gather for this study. The more models are included, the more complex the study. In this paper, we only compare two LES, as a trade-off to extract as much information as possible from a two-model comparison. We chose to collaborate with SwRI MRAMS LES for purely non-scientific reasons (mostly related to our contractor and funding requirements). It is clear that the OSU LES and MarsWRF LES models are equivalently entitled to be part of a future intercomparison effort.

We still believe that our two-model comparison, reinforced by a discussion on remaining challenges, paving the path towards a true intercomparison, provides a strong reference and guidance for futures intercomparisons of models and EDL studies, on Mars or other planetary environments. It shall be emphasized that, despite their heavy use to design EDL sequences of Martian missions, no intercomparison study of Martian atmospheric models is published. We therefore revised our paper accordingly to emphasize the necessary (yet still imperfect) first step we would like to impulse by comparing two LES models for Mars. We changed the title: "Comparison of two Large-Eddy Simulations of the Martian daytime convective boundary layer: sensitivity study and challenges." We also discuss this matter, along with other ideas/challenges, in the section added in the revised version and entitled: "Challenges of LES intercomparisons and suggestions for future studies".

**Section 5: I'm somewhat uncomfortable with this section in that you only perform sensitivity studies on one of the two models (the LMD model). This, then, becomes less of a model intercomparison and more of a sensitivity study of a single model. The two are quite different, and I would argue that the intercomparison study essentially ends in Section 4. I would like to see similar sensitivity studies for the SwRI model to evaluate whether, for example, the greater resolved TKE, or vertical wind speeds are more sensitive to parameter changes in the SwRI model than in the LMD model.**

We agree that the paper would benefit from such additional data. Nevertheless, the configuration of the project did not allow us to perform a sensitivity study with the two

models; it was only possible to carry out this study with the LMD model (this is also reflected in the authorship and the leadership of this study). Given those constraints, we had to make a choice of either include or not our sensitivity study in this paper. We decided to include it because

1. We explore the behavior of the simulated boundary layer convection to changes in settings, to explain the discrepancies observed between the two models. This further helps to identify the remaining challenges and directions towards which to point further work.

2. Such a sensitivity study using Martian LES has never been published before. We are convinced that our study should be of interest to many in the Martian community – and even the modeling community. This decision to include a sensitivity study is now reflected in the new title.

**Specific comments: Pg. 5, line 24: Can you expand on the validity of this assumption? In mesoscale modeling, there are 10s of km between the top of the 'good' results from the model, and the domain top. It seems that you're getting pretty close to the top of the domain when looking at the PBL, which comes in only a couple km below the model top. Are there issues with damping layers at the model top that might be affecting your results?**

The choice of the model top comes from previous simulations performed with the LMD LES (Spiga et al, QJRMS 2010: 145×145×201, 50 m LES with 12 km model top). It had been checked in these simulations that this choice did not impact the results. We performed an additional check for this paper by running our simulations with a much higher model top (16 km), along with higher horizontal simulation (see discussion below about domain size), and results slightly changed (PBL depth 25

**Figure 4: While you argue that the radiative forcing is about the same now after doing the radiative adjustment, the near-surface atmospheric temperatures are still vastly different–10 K in the nighttime and >20 K in the daytime based on**

**this figure. I can only imagine that is going to have a noticeable effect on the magnitude of turbulent activity at the smallest scales nearest to the surface. I don't see any discussion or acknowledgement of this difference. Surely it has to be important!**

The gap in the nighttime is actually less, as we can see on the new figure (the previous figure was biased because the levels used close to the surface were not the same than those in the SwRI model: this has been corrected). The daytime gap of near surface temperatures is, indeed, as much as 20 K. The near- surface schemes in the two models are very different and, despite our extensive efforts with matching the radiative heating rates, it has been impossible to fill this gap between the two SwRI and LMD 1D models with our methodology. We acknowledge this interesting difference in the 1D comparison and discuss it as a challenge in section 6.

**Pg. 11, line 28: You discuss 'quantitative discrepancies' between the models as being responsible for some of the differences between LMD and SwRI, and then refer forward to Section 6. I think it needs to be stated here what these quantitative differences are for the reader to understand and interpret the results of this section.**

Agreed. We modified the text to improve clarity.

**Pg. 14, line 2: To be honest, I don't think you've done any investigation of the discrepancies between the two models to this point. You've identified what they are, but you haven't done any interpretation of what is causing those discrepancies, or provided any insight into how they might be resolved.**

We agree with the reviewer. Now, interpreting the causes for discrepancies and providing insights into how they might be resolved is a very ambitious goal we are not able to completely fulfill with our study (although we have some clues, e.g. diffusive schemes, near surface scheme. . .). No studies published in the literature ever attempted to at least identify possible discrepancies between models sharing a common methological path (which was not the case in one of the only existing study comparing Martian mesoscale models, Kass et al. 2003). This is why we believe our study, imperfect as it is, forms a necessary first step to reach through future studies the overarching goal mentioned by the reviewer. We address this comment by changing the text at the beginning of section 5 and adding details in the section "Discussion".

**Pg. 16, line 21: This in an incomplete comparison–what were the model parameters in this 'other' LMD LES simulation? It's peculiar to say that the current results are comparable to past results that you don't show, because I, as a reader, have no objective way to assess that statement. What defines 'good agreement', for example?**

We changed the text: "The simulation remains typical compared to previous studies performed with the LMD LES. As an example, the PBL height obtained is in the 5-7 km range of what has been obtained for similar surface pressure values (see Spiga et al, 2010, Figure 2 case b and i)."

**Pg. 16, line 28: This is tied into the 'quantitative discrepancies' comment on Pg. 11, above. The difference in subgrid-scale diffusion scheme seems like a key difference that has gone unexplored. You've already gone to the effort to match as many physical parameters of the two models as you can, so why not the subgrid-scale diffusion scheme? It's somewhat of a cop-out to say that you see differences between the models, and then speculate on what might cause those differences (subgrid-scale diffusion) without trying to actually determine if it is, indeed, a cause. I think this study is incomplete because of this.**

See our previous comment. We agree our study is incomplete (although we prefer to qualify it as a "necessary first step"), but matching the subgrid-scale diffusion schemes between the two models is an ambitious work which would have required additional time, funding, and human resources to heavily modify the codes of one or the other model involved in our intercomparison. However, we completely agree this is an interesting and crucial point to address for future intercomparison studies. We discuss this point in the section "Challenges".

**Figure 5: Can you explain why there are far more points in the LMD curves than in the SwRI curves if both models have the same timestep? Is it as simple as more frequent output in LMD vs. SwRI? If so, why was that not coordinated? The higher frequency output of LMD gives the impression it is 'noisier' than SwRI, and it should probably be reduced to the same output frequency for plotting, if this is a rigorous intercomparison.**

Both simulations have the same physical timestep. It is indeed (the reviewer guessed right) a matter of output frequency. We corrected the figure so that both curves have the same output frequency. We deleted the figures showing the incident flux for sake of simplicity.

**Figures 10, 11, 12, 13, 16: Can you explain why in all of these figures, the SwRI LES output is truncated before the end of the time period under investigation? Also, in Figure 10, the LMD data is truncated at 17:00 as well. These figures need to be complete, or else an explanation given for their incompleteness. If it is due to something like a model crash, then this needs to be investigated and explained. I would not feel confident at all in model results that derived from a simulation that crashed. If it's just because the model was stopped because the interesting results had finished at a particular time, then this also needs to be explained and/or made consistent across all panels.**

The turbulent convection is active until the end of the afternoon (typically 16:30-17:00) and suddenly stops (the PBL collapses) when the surface becomes colder than the atmosphere above it (nighttime inversion). This occurs before 17:00 for some of the simulations. A comment has been added in the text.

**Typographical/minor issues: Figure 5 goes from 09:00-17:00. Figure 10 goes from 08:00-17:00. Figure 11 goes from 07:00-19:00. Figures 12 and 13 go from**
[Figure]

**11:00-17:00. Figures 16-21 go from 08:00-17:00. Why not plot everything on the same temporal axis? Consistency makes the reader happy, and the manuscript easier to follow**

OK this has been corrected for all figures. All other typographical/minor issues have been taken into account in the text. We carefully read the manuscript again to correct all typos.

Please also note the supplement to this comment:
http://www.geosci-model-dev-discuss.net/gmd-2016-241/gmd-2016-241-AC1-supplement.pdf

**Supplement:**

**A comparison of Large-Eddy Simulations of the Martian daytime convective boundary layer: sensitivity study and challenges**

Tanguy Bertrand[1], Aymeric Spiga[1], Scot Rafkin[2], Arnaud Colaitis[1], François Forget[1], and Ehouarn Millour[1]

[1]Laboratoire de Météorologie Dynamique (UPMC/CNRS), Paris, France
[2]Southwest Research Institute, Boulder, CO, USA

*Correspondence to:* Tanguy Bertrand (tanguy.bertrand -a- lmd.jussieu.fr)

[revised manuscript text omitted]

A systematic intercomparison between  all existing Martian LES models is still yet to be carried out to further characterize those differences.  Such an effort to compare all available models is beyond the scope (in term of human, time and funding resources) of the present study  (see section **??**). Here we propose, as a necessary first step

55 towards a true intercomparison, an unprecedented systematic comparison between two existing Martian LES models based on two distinct hydrodynamical solvers.

The need to evaluate the differences predicted by two distinct Martian LES is threefold:

1. Contrary to observational data, an estimate of the uncertainties of LES  diagnostics is still lacking. This is especially critical for the studies of atmospheric hazards during EDL, given the central role that LES  diagnostics play in those studies and the relative paucity of available data to characterize the Martian PBL dynamics (e.g. for vertical winds). Comparing LES models would provide guidance on the range of model variance (i.e. the spread in modeling results for a similar Martian site and season), thereby enabling an optimal EDL design for both landing spacecraft and definition of landing ellipse.

2. Carrying out a LES  comparison would highlight discrepancies between results and help to identify the specific areas in which model improvements would be the most helpful. This overarching goal is beneficial for the whole Martian science. For instance, turbulent wind variability (i.e. "gustiness") plays an important role in controlling dust lifting on Mars (e.g., **?**, and references therein). Since turbulent wind measurements on Mars are very incomplete, LES predictions are still being an important source to assess the wind conditions on Mars associated with dust lifting (**?**). More generally, the continued development of Martian LES models is also of crucial importance to better understand the mechanisms responsible for heat and momentum transfer both by daytime PBL mixing and surface-atmosphere interactions. Following the tendency drawn by terrestrial studies, Martian LES predictions are more and more used to build and improve PBL parameterizations in Global Climate Models (GCMs) for Mars (**?**).

3. Since Martian LES rely on hydrodynamical solvers inherited from terrestrial studies, confronting those models to the intense PBL convection on Mars (compared to the Earth, cf. **?**) provides a stringent test for those solvers.  A comparison study of two or ideally more Martian LES will ultimately be a strong driver of improvement for the atmospheric models used to carry out LES on Earth, and in an increasingly diverse range of planetary conditions (e.g. Venus LES, **???**).

In this paper, we compare the LES results obtained by, on the one hand, the Laboratoire de Météorologie Dynamique (LMD) Martian mesoscale model (**??**) and, on the other hand, the  Southwest Research Institute (SwRI) Martian mesoscale model (**??**). We develop a strategy which makes our  comparison study the first one of its kind for Mars: we ensure that similar physical constants and radiative forcing are employed in both models before performing a comparative analysis of LES results and conclude on the performance of the two dynamical solvers in predicting Mars' PBL convective motions. We further complement this  comparison study by an exploration of the sensitivity of the convective PBL predicted by the LMD LES to surface thermophysical properties (e.g. albedo), ambient wind, and atmospheric dust loading.

 The present LES comparison has been performed in the context of the European Space Agency (ESA) ExoMars 2016 mission (hereinafter referred as ExoMars) with the aim of providing constraints for the EDL of the ExoMars Demonstrator Module (EDM, also named Schiaparelli). LES modeling  has therefore been performed at the ExoMars landing site, namely in the Terra Meridiani region (latitude $-1.82°$N, longitude $-6.15°$E),  at the landing date in northern autumn (solar longitude $L_s = 244°$).

In  section **??**, we describe the LMD and SwRI models used in this LES comparison. In section **??**, we provide details on the  comparison strategy, and how we reached similar radiative forcing both in SwRI and LMD LES. The results of both the Martian LES  comparison and sensitivity study are discussed in section **??** and **??** respectively. Finally, in section **??**, we discuss the challenges of LES intercomparison and we suggest a possible path forward for future LES studies.

**2 Models description**

We provide here the key points to describe the two models used for our LES comparison. Further details about each model can be found in the references provided in this section.

Both LMD and SwRI Martian LES models have been built independently by adapting terrestrial mesoscale models to the Martian case, with the coupling of specific physical models (namely, radiative transfer and soil model) initially developed for Martian GCMs:

- LMD LES are performed using the LMD Martian Mesoscale Model (**??**), based on the Weather Research and Forecast (WRF) model and its fully compressible non-hydrostatic dynamical core (**??**), combined with the comprehensive set of physical parametrizations of the LMD GCM (**?**).

- SwRI LES are performed using the Mars Regional Atmospheric Modeling System (MRAMS), a nonhydrostatic Martian mesoscale model developed at SwRI (**??**) and based on the terrestrial RAMS dynamical core (**?**), in which physical parameterizations are inherited from the Martian NASA Ames GCM (**?**).

The two LES models not only use very distinct different radiative transfer and soil  models (inherited from GCM), but also use different dust scattering properties (at the time the runs for this study were carried out, **?** for SwRI LES vs. **?** for LMD LES), which can lead to significant departures in the predictions of atmospheric temperatures.

[revised manuscript text omitted]

SwRI models. We chose to vary 1. the dust extinction efficiency $Q_{ext}$ (or "thermal infrared opacity", at the reference infrared wavelength) and 2. the visible single scattering albedo $w_0$, which quantifies dust "brightness" (further details and explanations in **??**). For the sake of simplicity, we define the ratios $S$ and $C$ of the corrected value over the reference value for respectively $w_0$ and $Q_{ext}$ in the LMD model. The higher $S$, the brighter dust; the higher $C$, the stronger absorption by dust in the infrared.

**??** shows the sensitivity of the temperature profiles to $S$ and $C$. Here, the comparison of profiles is focused above the PBL because in 1D, the PBL is parameterized while it is resolved in LES (see next section for comparisons within the PBL). We performed a comparative study for all local times, given that LES runs span stability conditions from highly unstable around noon to highly stable in early morning and late afternoon. For the sake of illustration, we show in the aforementioned figures typical nighttime and daytime temperature profiles at local times $0:00$ and $12:00$. In the clear atmosphere case we adopted (setting dust opacity to $0.2$), dust brightness $S$ has a strong impact on the temperature profiles: a decrease of $20$ K is obtained in nearly the whole atmospheric column when increasing $S$ from $0.92$ to $1.04$. Indeed bright dust means higher dust albedo, hence increased reflection of solar flux to space, and colder atmosphere and surface. The dust extinction ratio $C$ has a lesser impact (though not negligible) than $S$ on the thermal structure, as shown by **??**. This is expected for a clear atmosphere, where dust absorption in the infrared is not significant enough to cause a strong heating of the atmosphere (contrary to dust storm conditions). This sensitivity study shows that the values of the dust visible scattering albedo and extinction efficiency should be decreased to simulate the effect of dust aerosols both darker and more strongly absorbing in the infrared. The effect of this correction is to warm the atmosphere (both during daytime and nighttime).

As the tuning of these dust parameters does not change significantly the surface temperatures (because of the relatively clear atmosphere considered), the surface thermal inertia has also been changed in the LMD model so that it replicates the SwRI diurnal cycle, at least during daytime when the surface temperatures impact the PBL. Many optimization loops have been performed to replicate in an optimal way the  SwRI temperatures at all local times, taking into account

[Figure]

**Figure 4.** Same as **??** with $S = 94\%$ and $C = 70\%$ in the LMD 1D model.

the change of surface temperature as well.  Values of $S = 94\%$ and $C = 70\%$ are used, in order to bridge the $5 - 10$ K gap between the LMD and SwRI temperature profiles ( **??**). In addition, by increasing the thermal inertia up to 300 tiu (which remains close to the surface conditions encountered at ExoMars landing site), we decrease the LMD surface temperatures  by about 8 K around noon and replicate the SwRI daytime surface temperatures, as shown in  section **??** by **??**.

However, there is still a 20 K gap in the daytime near-surface temperatures that cannot be reduced (over a range of plausible parameters) by optimizing the radiative forcing as it is the case for the rest of the PBL. This suggests that differences in the formulation of surface layer schemes in the two models (see (**?**) for MRAMS and (**?**) for LMD LES) might explain such a difference. This remains to be explored, as discussed in section **??**.

**4  Comparison of LES results**

This section compares the LMD and SwRI LES results, obtained using the settings described by Table **??** and Table **??**, with the modified radiative transfer properties detailed in section **??** which allow a similar radiative forcing between the two models ($S = 94\%$, $C = 70\%$ and thermal inertia = 300 tiu). Two main test cases are considered: LES devoid of any ambient wind, and LES with an ambient wind of 15 m.s$^{-1}$, thereby providing typical LES results of Martian daytime boundary convection in windless and windy conditions at the ExoMars landing site.

**4.1  Forcing of PBL activity**

The PBL activity and its evolution are controlled by the surface temperature and the surface incident solar flux. As described in section **??**, the preliminary steps of convergence of both models enabled a similar forcing to be reached, as shown by **??**. During daytime, discrepancies between LMD and SwRI LES surface temperatures are less than 1 K, and we checked that the shortwave and longwave incident solar flux are similar at all local times. In addition, we  checked that the atmosphere above the PBL

265 is controlled by similar radiative forcing in both models, as is found in the previous section with the 1D models. With these settings, any difference observed in the LES results (PBL depth, maximum updrafts...)  should not be related to differences in "background" radiative thermal structure, but rather to different dynamical approaches in both models.  Other possible sources of differences are discussed in section ??. For example, as mentioned above, the two models use different PBL  surface layer schemes, which  can impact the results (see  sections ?? and

270 ??).

**4.2 Comparison of typical LES wind structures**

In this section we analyze the turbulent wind structures in order to verify if they are similar in both models and consistent with typical LES diagnostics. Note that the ambient wind is westward in the simulations.

**4.2.1 Wind structures and small scale variability**

275 ?? shows horizontal sections of vertical winds at about 250 m and 2 km above the surface and at local time 11:00, obtained without ambient wind. Both models predict a horizontal organization of vertical winds into polygonal cells with narrow updrafts on the ridges of the cells and large subsidence in the middle of the cells, with same spatial dimensions and structure sizes. These are a typical pattern of the PBL which remain in conformity with typical LES studies. At 250 m altitude, maximum updrafts and downdrafts reach up to 8 m.s$^{-1}$ and -6 m.s$^{-1}$ in both models, with slightly higher values over few grid points in SwRI

280 results. At 2 km altitude, where updrafts are more vigorous, the differences between both results are more pronounced. LMD LES  predicts a maximum updraft value of 8 m.s$^{-1}$ while SwRI LES results show values twice higher, around 15 m.s$^{-1}$ (see  section ??). Because of mass conservation, downdrafts are also found to be more vigorous in the SwRI LES than in the LMD LES.

?? and ?? show horizontal sections of horizontal wind amplitudes at local time 11:00, at about 5 and 250 m above the
285 surface, in the case of an ambient wind of 0 and 15 m.s$^{-1}$ respectively.

In the case without ambient wind, horizontal wind structures are similar in both models and typically show gusts in all directions related to the convective cells in the PBL. However, it can be noticed that the variability at very small scale (i.e. a few grid points apart) is much higher in SwRI results than in LMD results. This higher small-scale variability appears in all SwRI simulations . They may be related to the quantitative discrepancies  seen in PBL diagnostics
290 (section ??) and summarized in Tables ??-??. This is further discussed in section ??. Here, maximum horizontal wind amplitudes at 5 m above surface are found to be in the range 6-8 m.s$^{-1}$ in both LMD and SwRI LES, although SwRI results also show maximum values at 12 m.s$^{-1}$ over a few grid points. At 250 m above surface, horizontal winds reach 6 m.s$^{-1}$ in both models (9 m.s$^{-1}$ over a few grid points in the SwRI model).

When a sufficiently strong ambient wind is added (typically more than 10 m.s$^{-1}$), it distorts the organization in polygonal
295 cells and forces the gusts to propagate in the same direction, as shown by ??. In this case, LMD wind amplitudes are found to be larger than SwRI values, especially 250 m above the surface, although the wind structures remain similar in both LES. Maximum values for horizontal wind are 16-18 m.s$^{-1}$ in both LES.

This difference in altitude is highlighted by **??**, showing the mean profile of horizontal winds in both models between 11:00 and 12: 00, in the simulation with 15 m.s$^{-1}$ ambient wind. SwRI results show weaker horizontal winds than LMD results in
300   the whole PBL. Although these differences could be related to the different intensity of the convection, they may also be related to the different scheme of near-surface stability. The large differences encountered in the first hundreds of meters reinforce this possibility  (see section **??**). It can also be noted that SwRI results show a deeper mixed layer, which is consistent with the overall weaker winds (momentum is mixed through a deeper layer).

**4.3 Comparisons of PBL diagnostics**

305   LES results compared in this section are the PBL depth, the turbulent kinetic energy (TKE) and the maximum updrafts and downdrafts within the PBL, which are key characteristics of the convection inside the Martian PBL. The PBL depth is directly and positively correlated to the intensity of its dynamics while the TKE is a measure of turbulence intensity and is directly related to the transport of heat and momentum through the PBL. An increase of TKE denotes a more turbulent PBL. Updrafts and downdrafts also enable  the comparison of the intensity of the PBL turbulence in both models. Table **??** and
310   Table **??** summarize the results.

**4.3.1 Planetary boundary layer height**

**??** shows that the daytime PBL depth grows faster and reaches a higher vertical extent in the SwRI LES than in the LMD LES. By the end of the afternoon, the PBL depth reaches 7.5 km (SwRI) and 4.8 km (LMD) without ambient wind and slightly more in windy conditions. These results confirm that the convection in the SwRI LES is more vigorous. In fact, the
315   maximum PBL depth obtained in the SwRI LES is close to the highest values that could be found on Mars and measured through  radio occultation, which is about 8-10 km (**?**). It should be pointed out, however, that these simulations contain none of the realistic physics that would tend to suppress a PBL, notably large-scale subsidence or regional circulations (e.g. as experienced in Gale Crater, see **?**). The turbulent convection is active until the end of the afternoon (typically 16:30-17:00) and suddenly stops (the PBL collapses) when the surface becomes colder than the atmosphere above it
320   (nighttime inversion). For some of the simulations, this occurs before 17:00, explaining the sharp drop in **??**.

**4.3.2 Turbulent kinetic energy**

The differences in the vertical extent of PBL mixing between both LES can also be inferred from the variations of (resolved) TKE, as shown by **??**. Although the convective activity rapidly declines at same local time 17:00 and the maximum TKE values occur around same local times 13:00-14:00 in both models, larger TKE values are found in the SwRI LES, in consistency
325   with previous results,. Without ambient wind, the maximum TKE in the PBL reaches 12 m$^2$.s$^{-2}$ around 1.5 km altitude in LMD results, while it reaches about 21 m$^2$.s$^{-2}$ around 3.5 km altitude in SwRI results, a factor of two more intense. Similar differences are found in the LES with windy conditions.

| LES without ambient wind | LMD | SwRI | Difference (%) |
|---|---|---|---|
| PBL  height (km) | 4.8 | 7.5 | 56 |
| Turbulent  heat flux (K.m.s$^{-1}$) | < 1.25 | < 3.2 | 156 |
| Turbulent  kinetic energy (m$^2$.s$^{-2}$) | < 12 | < 21 | 75 |
| Maximum  updraft speed (m.s$^{-1}$) | < 15 | < 24 | 60 |
| Maximum  downdraft speed (m.s$^{-1}$) | < 8 | < 12 | 50 |

**Table 3.** Maximum PBL diagnostics values from the LMD and SwRI LES without ambient wind

**4.3.3 Maximum speed for convective updrafts**

In line with previous results, **??** shows that maximum updraft speeds obtained from SwRI LES are higher than those from LMD LES, with a ratio of about 1.5-1.8, all over the mixing layer depth between local times 11:00 and 17:00. In LES without ambient wind, maximum speeds reach 13-15 m.s$^{-1}$ in LMD results around local times 12:00 – 14:00 between altitudes 2 to 4 km while they reach 20-24 m.s$^{-1}$ in SwRI results between altitudes 2 to 6 km at same local times. In windy conditions, maximum updraft speeds in both LMD and SwRI LES remain in similar ranges. This is expected given that SwRI has finer structure (narrower updrafts, presumably less diffusive).

**4.3.4 Maximum speed for convective downdrafts**

In both LES, maximum updraft speeds (**??**) are in comparison larger by a factor 2 than maximum downdraft speeds (**??**). This is a consequence of the organization of turbulence in cells with narrow updrafts and broader downdrafts. Both LES predict maximum downdraft between local times 13:00 and 15:00. Without ambient wind, downdraft up to 8 m.s$^{-1}$ are predicted by the LMD LES from 200 m to 5 km above the surface while the SwRI LES predicts values up to 12 m.s$^{-1}$. In the LES with ambient wind, these values slightly increase to 9 m.s$^{-1}$ and 15 m.s$^{-1}$ respectively.

**4.3.5 Distributions of vertical wind speeds**

**??** and **??** show the distribution of vertical wind speeds obtained between local times 13:00-14:00 and altitudes 250-5000 m, for both LES without and with ambient wind respectively. In the windless case, 95% of vertical wind speeds are in the [-6,6] m.s$^{-1}$ range for LMD and [-9,9] m.s$^{-1}$ for SwRI. In the windy case, 95% of vertical wind speeds are in the [-7,7] m.s$^{-1}$ range for LMD, and [-11,11] m.s$^{-1}$ for SwRI. Therefore, the strongest vertical winds represent a very low probability. As an example, in the SwRI LES, both the maximum downdraft value of 11 m.s$^{-1}$ and the maximum updraft value of 28 m.s$^{-1}$ represent less than 0.01% of all values.

| LES with 15 m.s$^{-1}$ ambient wind | LMD | SwRI | Difference (%) |
|---|---|---|---|
| PBL  height (km) | 5.3 | 8.1 | 53 |
| Turbulent  heat flux (K.m.s$^{-1}$) | < 1.7 | < 3.4 | 100 |
| Turbulent  kinetic energy (m$^2$.s$^{-2}$) | < 15 | < 34 | 126 |
| Maximum  updraft speed (m.s$^{-1}$) | < 15 | < 22 | 46 |
| Maximum  downdraft speed (m.s$^{-1}$) | < 9 | < 15 | 60 |

**Table 4.** Maximum PBL diagnostics values from the LMD and SwRI LES with 15 m.s$^{-1}$ ambient wind

**5 Sensitivity simulations**

This section presents a sensitivity study of the Martian daytime PBL properties in the LMD LES model to dust loading, surface
albedo, ambient wind and subgrid scale diffusion coefficient, using the same reference settings as described in  section
**??**, without ambient wind. This enables to  explore the role of forcing  of the Martian
PBL  and to assess if an uncertainty of
one parameter can explain the discrepancies evidenced between the LMD and MRAMS LES results. Furthermore,
 such a sensitivity study has seldom been detailed in the existing literature.

**5.1**

**5.1 Sensitivity to dust opacity**

LES have been performed using distinct dust optical depths of 0.2 (reference), 0.6, 1 and 3. Comparing LES predictions in those
three cases reveals dramatic differences in the strength of the boundary layer convection. **??** shows how the daytime evolution
of boundary layer depth is influenced by dustiness in the Martian atmosphere. Firstly, the mixing layer is of significantly lower
vertical extent for higher dust opacities (which is true during the whole day). At local time 14:00, boundary layer depths are
respectively 3.8, 3, 2.5, 1.7 km for dust loadings 0.2, 0.6, 1, 3. This behaviour originates from dust absorption of solar radiation
in the visible leading to decrease in surface temperature. Moreover, stability in the lowest layers of the atmosphere is enhanced
by dust heating, which leads to a less vigorous boundary layer turbulent convection.

Secondly, the PBL is collapsing earlier in the afternoon when dust opacity increases: while the PBL depth is still maximum at local time 17:00 in a clear atmosphere, it starts to slightly decay for a dust opacity at 0.6 and more distinctively decrease for an opacity at 1 (the maximum vertical extent in this case is attained around local time 15:30). The most extreme case (dust opacity at 3) shows a very limited growth of the convective boundary layer, which peaks at the very low value (with respect to Martian standards) of 1.7 km between local times 13:00 and 14:00, before a rapid collapse of the turbulent convection occurs at local time 14:00-14:30.

Both the limitation of convective activity and the displacement of its maximum towards earlier afternoon associated with an increase in dust optical depth can also be assessed by the comparison of maximum (resolved) TKE shown on **??**. Differences in TKE between the clear and extremely dusty cases are about one order-of-magnitude – even differences between the opacity 0.2 and 1 are significant (50 to 60% decrease). It is interesting to note that, in theory, dust radiative heating by absorption in the visible (plus a smaller contribution in the infrared) should cause TKE and updraft speeds to increase with dust opacity. Present LES results with complete radiative transfer show this effect does not significantly compensate the aforementioned influence of dust loading on surface temperature and atmospheric stability. Overall, the dustiness of the Martian atmosphere strongly determines the strength of boundary layer convection and this has important consequences on conditions for EDL systems as ExoMars landing between 14:00 and 16:00. This is exemplified by maximum vertical wind speeds (not shown). Maximum updraft and downdraft values throughout the whole day vary dramatically with dust opacity: updrafts of 15, 12, 10, 4 m.s$^{-1}$ and downdrafts of -9, -7, -5, -2 m.s$^{-1}$ are predicted by LMD LES for dust opacities 0.2, 0.6, 1 and 3. Again, convection for an opacity at 3 is severely limited. Vertical winds are lower at 16:00 than at 14:00 in all cases, but the differences are more prominent between the two local times when more dust is suspended in the Martian atmosphere.

Finally, it is important to note that if the dust opacity values follow a geometric progression which implies a linear increase in atmospheric heating, variations of PBL depth, TKE and maximum winds with dust opacity are not linear. The sensitivity of PBL turbulence to dust opacity is a complex combination of dust influencing surface temperature, atmospheric stability and maximum turbulent heat flux as well as turbulent motions adjusting to those various modified forcings. Only LES employing full radiative transfer can address this complexity.

**5.2   Sensitivity to surface albedo**

In order to assess the sensitivity of the PBL to surface conditions, the LMD LES has been tested with surface albedo value of 0.1, 0.21 (reference), 0.4 and 0.6. A value of 0.1 is extreme, although not unrealistic, and causes surface temperatures to be significantly warmer, hence boundary layer convection to be more vigorous, as shown by **??** and **??**. A lower albedo not only  causes the sensible heat flux to be larger, but also the infrared radiative flux emitted by the surface and absorbed by $CO_2$ and dust in the lowermost atmospheric layers to be larger. The PBL depth predicted by LES is about 20% higher for the 0.1 albedo case, and 35% lower for the 0.4 case, compared to the 0.21 case. Similar conclusions apply for the maximum TKE predicted. The PBL convection on a more reflective surface is very limited, with a height of 1.5 km for an albedo up to 0.6.

**5.3 Sensitivity to ambient wind**

The sensitivity of the PBL to the ambient wind is addressed by running the LMD LES with 0, 15 (both reference runs) and 25 m.s$^{-1}$ ambient wind. In the afternoon, the wind enhances convection by increasing the surface-atmosphere heat and momentum transfer in the surface layer. Larger wind speeds thus yield higher values for turbulent heat flux, and higher values of TKE (according to the TKE equation). **??** shows that the PBL in windy conditions is especially vigorous between local times 12:00 and 15:30. In addition, the boundary layer convection appears to start earlier in the morning in windy conditions. The maximum TKE predicted by LES in the afternoon is about 20% higher for the 15 m.s$^{-1}$ case (40% for the 25 m.s$^{-1}$ case) compared to the no-wind case. In such conditions, the maximum PBL depth is higher by about 500 m (10%) in the 15 m.s$^{-1}$ case and 700 m (15%) in the 25 m.s$^{-1}$ case, as shown by **??**. Quantitative estimates about maximum vertical winds have to be raised to about 10% and 15% in the 15 m.s$^{-1}$ and 25 m.s$^{-1}$ ambient wind cases respectively, compared to the no-wind estimates (figure not shown).

**6**

**5.1 Sensitivity to subgrid scale diffusion**

Can a difference in the small-scale diffusion schemes in both models be related to the difference of values of PBL depth, maximum TKE and vertical wind speeds detailed in section **??**? In an attempt to answer this question, the LMD LES model has been run with different subgrid scale diffusion coefficients (or mixing coefficients). Tested with much lower values than the typical one used (0.15 at LMD, decreased by a factor up to $10^4$), the PBL diagnostics remain similar in the afternoon (**??**). The same conclusions apply when analyzing the maximum TKE or wind speeds, confirming that the small-scale mixing coefficient has a negligible impact on the development of the convective PBL in the daytime. This exploration is incomplete because the discretization of primitive equations in dynamical core may cause the dynamical core to be naturally diffusive (see section **??** for further details), which cannot be controlled by the above-mentioned subgrid-scale mixing coefficients.

**5.2 Sensitivity to domain size**

The  LES results can be sensitive to domain size, as was described in section **??**. We ran the model with a much wider domain defined by a 250x250x250 grid, and a model top at 16 km instead of 12 km (horizontal and vertical resolutions remain the same than in the reference case). Results show an increase of the PBL depth of 20-25% (**??**), which is not sufficient to explain the 100% differences observed between both LMD and SwRI LES – although it might explain part of the discrepancy.

**6 Challenges of LES intercomparison and suggestions for future studies**

This section aims to identify the key challenges of LES intercomparisons and gives suggestions for a possible path forward for future LES studies.

A first difficulty is to determine which models shall be selected to be intercompared. Ideally, the best intercomparison would include all existing models of the community. However, the more models are included, the more multi-dimensional the intercomparison study becomes, with exponential human, time and computing resources needed. Furthermore, the fact that the models have been built independently may lead to several issues: a specific parameter to be explored in the intercomparison may not exist, or may not be easily accessed in all of them; the models have their own strategy for numerical stability, and imposing a unique setting for all the involved models might be detrimental to the quality of their diagnostics; the models use different options, inputs, and outputs (their initialization might actually be a problem on its own within the intercomparison project); the vertical grid of the models may differ, with heights fixed above ground in a sigma-z formulation much different than the sigma-p coordinate used in other models (possibly leading to spurious effects related to one or the other choices for vertical coordinates). A straightforward solution could be to first compare models sharing the same dynamical core, in order to identify any difference related to the physical packages (radiative transfer, mixing, etc...), and then extend the intercomparison to models built on distinct dynamical cores.

One of the difficulties encountered in this paper is the matching of the near-surface temperatures. Surface layer schemes in respectively the LMD and SwRI LES models are based on distinct formulations; we suspect the impossibility to match near-surface temperatures between the two models with our radiative transfer explorations, which suggests that the impact of those differences in surface layer schemes on the near-surface temperature structure in LES is significant and deserves to be explored by a dedicated study. This impact is difficult to evaluate: it affects the magnitude of turbulent activity near the surface, which in turn modifies the sensible heat flux (through the surface layer schemes), which in turn changes the near-surface temperature profile. Future intercomparison studies should acknowledge this issue and try to converge towards similar near-surface temperature profiles.

Another key difference which has been found between the LMD and SwRI  models is related to subgrid-scale diffusion. This difference of diffusion is suspected to significantly impact the intensity of the convection within the PBL. Consequently, matching (or, if ever possible, deactivation) of subgrid-scale diffusion schemes between models is an important task to realize consistent intercomparisons. This task is more ambitious than it seems. One difficulty is that subgrid-scale numerical diffusion does not depend on only one parameter. For instance, the LMD LES is based on the WRF dynamical core which is inherently diffusive owing to the chosen discretization of the primitive equations (a diffusive term is added for odd-order advection operators (**?**)). Thus, it is not trivial to completely disable subgrid-scale mixing in a model – in this paper, we tested the LMD LES model sensitivity to the subgrid mixing coefficient, but it has little effect on the results since diffusion terms remain inherent to the formulation of the dynamical core (see **??**). Future LES intercomparisons should further investigate the impact of the subgrid-scale mixing on Martian LES results, which appears of central importance in LES and mesoscale models, and probably even in Global Climate Models. This question has remained eluded in most Martian modeling studies to date.

465     Another challenge for Martian LES is the size of the modeling domain. As stated by **?**, the horizontal size of the domain must be large enough so that the periodic boundary conditions cannot influence the turbulence computed for the domain interior. Generally speaking, the length of the grid should be three times the size of the largest eddy that will be resolved by the simulation. Since Martian PBL depth typically reach 10 km, the grid size should be around 30 km. In addition, in order to resolve the smaller eddies, the horizontal resolution must be fine enough, typically around tens of meters. Consequently,

470 for a 30 km square LES domain with a 50 m gridspacing, the number of computational locations would then be 600x600. This is challenging to achieve, although largely within reach of modern supercomputers as is demonstrated by a recent study about statistics of dust devils in Martian LES (**?**). In this paper, the model sensitivity to the size of the domain has been tested and results show an increase of PBL depth about 20-25% (see **??**), in line with previous LMD LES studies ((**?**)). Future intercomparisons should be careful about the domain size and top, and test their model with different configurations.

475     Finally, regarding the strategy we adopted to avoid a comparison of radiative schemes, one suggestion for future Martian LES intercomparisons would be to compare the models with no dust loading at all. This has not been considered in this paper, because the context of the ExoMars mission required LES to be carried out with the most realistic possible temperature profile. LES intercomparisons with a dust-free atmosphere would be a good starting point for future studies – aimed at theoretical discussions and not EDL discussions as in the present paper, since it would enable to compare surface temperatures without

480 the complication of the radiative properties of airborne dust.

**7   Discussion**

The steps of the comparison between the LMD and SwRI LES models can be summarized as follows: LES have been performed at ExoMars landing site and date using settings as similar as possible. A tuning of the LMD radiative transfer routine has been necessary to ensure a similar radiative response of both LMD and SwRI models to same forcings. This tuning involved slight

485 changes of dust properties (extinction and brightness) and surface thermal inertia (increased from 238 tiu to 300 tiu in the LMD LES), essential to obtain similar daytime surface temperatures in both LES and thus to similarly force the turbulence within the PBL.

    The comparison of LES shows similar qualitative results (vertical wind organized into polygonal cells, horizontal gusts) but different quantitative results. SwRI results show values of heat flux, kinetic energy, updraft  and downdraft speeds which

490 are more dispersed, with maximum values higher than LMD results with a ratio between 1.5 and 2, as summarized by Table **??** and Table **??**. This leads to an almost twice more vigorous PBL in the SwRI LES than in the LMD PBL, even though the maximum values only represent a very small fraction of the domain. Results remain similar with or without ambient wind.

    It is important to note that all the results and values obtained from both models remain realistic (with the caveat in mind that no measurements of vertical wind in the Martian convective boundary layer are available from previous missions).

495

    The simulations performed with the LMD LES remain typical compared to previous studies performed with the same model. As an example, the PBL height obtained is in the 5-7 km range of what has been obtained in previous studies in conditions

close to the ExoMars landing site  (e.g similar surface pressure values, see **?**, figure 2 cases b and i). In addition, these values are consistent with radio-occultation measurements (**?**).The boundary layer depths predicted by the SwRI LES model, although clearly in the upper range, are still consistent with those measurements.

The discrepancies observed between both LES  cannot be explained by differences in boundary conditions or radiative forcing, which are similar in both models (less than 1 K difference). Identifying exactly the origin of those discrepancies is challenging (see section **??**) and beyond the scope of the present study, which is only a first step toward an intercomparison of Martian mesoscale and LES models. In this study, we identified the discrepancies between two Martian LES and drew possible future areas of research to disentangle the causes underlying those discrepancies. It has been found in  section **??** that SwRI results exhibit a higher variability at a very small scale than LMD results. This could stem from different assumptions in the subgrid-scale diffusion schemes adopted in both LES. On the one hand, the SwRI LES reflects a much weaker subgrid-scale diffusion than the LMD LES, which would put the SwRI LES at greater risk to overestimate maximum vertical winds ("noisy" turbulent signals). On the other hand, the LMD LES appears to have a subgrid-scale diffusion which efficiently removes the accumulation of energy at the grid point scale, although the possibility still exists that this subgrid-scale diffusion might be too strong, yielding underestimated vertical winds. We tested different subgrid scale diffusion coefficients with the LMD LES model but these changes did not significantly affect the PBL. Consequently, it is plausible that the differences observed between both models lie deeper in the assumptions of their small-scale diffusion schemes, in the inherent diffusion within the dynamical cores (e.g., the advection operators and possibly other numerical diffusion assocated with different schemes) or in other dynamic parts of the models (e.g.  distinct discretizations of the hydrodynamical equations). This remains to be explored further in the respective models. Finally, as suggested in  section **??** and **??**, differences in the PBL near-surface scheme used in both models could also impact the LES results.

 Results in section **??** show that the PBL can be strongly affected by large changes of dust loading and surface conditions. In contrast, although the influence of ambient wind on the Martian PBL turbulence is a significant component to be taken into account for EDL studies, results with ambient wind will not change drastically compared to no-wind simulations (here the Martian situation is quite different from the Earth due to the radiative control of the boundary layer). Comparing windy simulations (**??**, **??**) with extreme soil simulations (**??**, **??**) or with dusty simulations (**??**, **??**) shows that windy conditions represent a secondary influence of Martian PBL convection and  are likely to be overcome by changes in the primary forcing of the Martian PBL, that is, radiative control. Turbulent convective activity is enhanced in windy conditions, but this is overwhelmed by the strong suppression of boundary layer growth caused by very dusty conditions.

The conclusions of the  comparison campaign presented in this paper do not prevent LES from being relevant tools to study the PBL turbulence on Mars and to provide constraints to assess atmospheric hazards encountered by future landing systems, provided caution is exerted along the quantitative lines drawn by the estimates in this paper. Above all, improving the diagnostics provided by Martian LES will require more complete observations of the Martian PBL turbulence in future in situ missions to Mars.

**8 Data availability**

The SwRI and LMD files, figure data or any other source data of this paper are freely available upon request by contacting T.B. or A.S. (tanguy.bertrand@lmd.jussieu.fr, spiga@lmd.jussieu.fr).

535 *Author contributions.* Author contribution

T.B.  and A.C. prepared the comparison of models. T.B. performed the LMD simulations, S.R performed the SwRI simulations. Both T.B. and A.S  wrote the manuscript, with contributions from S.R., E.M. and F.F.

*Competing interests.* Competing interests

540 The authors declare no competing financial interests.

[Figure]

**Figure 5.**  Surface temperatures in SwRI (black) and LMD (red) LES models.

[Figure]

**Figure 6.** LMD (left) and SwRI (right) horizontal section of vertical velocity at about 250 m (top) and at about 2 km (bottom) above the surface, at local time 11:00. Simulations without ambient wind.

[Figure]

**Figure 7.** LMD (left) and SwRI (right) horizontal section of horizontal wind amplitudes at first level above surface (top) and at about 250 m above surface (bottom), at local time 11:00. Simulation without ambient wind

[Figure]

**Figure 8.** LMD (left) and SwRI (right) horizontal section of horizontal wind amplitudes at first level above surface (top) and at about 250 m above surface (bottom), at local time 11:00. Simulation with 15 m.s$^{-1}$ westward ambient wind.

[Figure]

**Figure 9.** Horizontal wind profile in the SwRI (black) and LMD (red) LES models, between local times 11:00-12:00 in the simulation with 15 m.s$^{-1}$ ambient wind.

[Figure]

**Figure 10.** LMD and SwRI variations of boundary layer depth between local times 0908:00 and 17:00 and altitudes above ground 0 and 9 km. Simulations without ambient wind and with 15 m.s$^{-1}$ ambient wind.

[Figure]

**Figure 11.** LMD (left) and SwRI (right) variations of turbulent kinetic energy between local times 07:00 and 19:00 and altitudes above ground 0 and 9 km. Simulations without ambient wind (top) and with 15 m.s$^{-1}$ ambient wind (bottom).

[Figure]

**Figure 12.** LMD (left) and SwRI (right) maximum speeds for convective updrafts reached in the simulation domain between local times 11:00 and 17:00 and altitudes above ground 0 and 8 km. Simulations without ambient wind (top) and with 15 m.s$^{-1}$ ambient wind (bottom).

[Figure]

**Figure 13.** LMD (left) and SwRI (right) maximum speeds for convective downdrafts reached in the simulation domain between local times 11:00 and 17:00 and altitudes above ground 0 and 8 km. Simulations without ambient wind (top) and with 15 m.s$^{-1}$ ambient wind (bottom).

[Figure]

**Figure 14.** Histogram of vertical wind speeds for local times 13:00-14:00 and altitudes 250-5000 m. LMD results are in green, SwRI results are in blue. No ambient wind.

[Figure]

**Figure 15.** Histogram of vertical wind speeds for local times 13:00-14:00 and altitudes 250-5000 m. LMD results are in green, SwRI results are in blue. With 15 m.s$^{-1}$ ambient wind.

[Figure]

**Figure 16.** Evolution of the boundary layer depth for different dust optical depths (LMD LES)

[Figure]

**Figure 17.** Evolution of the maximum turbulent kinetic energy for different dust optical depths (LMD LES)

[Figure]

**Figure 18.** Evolution of the boundary layer depth for different surface albedo values (LMD LES)

[Figure]

**Figure 19.** Evolution of the maximum turbulent kinetic energy for different surface albedo values (LMD LES)

[Figure]

**Figure 20.** Evolution of the maximum turbulent kinetic energy for different large scale ambient winds (LMD LES)

[Figure]

**Figure 21.** Evolution of the boundary layer depth for different large scale ambient winds (LMD LES)

[Figure]

**Figure 22.** Evolution of the  boundary layer depth for different  subgrid scale  mixing coefficient and a case of larger domain (LMD LES). The fist part of the curve of the larger domain simulation is missing because of technical issues but this does not alter the comparison made in the afternoon.

---

## Author Comment (AC2) · 31 Mar 2017

Dear Dan,

Many thanks for your review and insightful comments. We believe that those comments complements very well our manuscript by helping to identify the many challenges of model intercomparisons, an overarching goal truly difficult to address. Probably our submitted version reads as the definitive work on this topic, while this was absolutely not our intent. The goal of our paper is to provide the community with the report of a first attempt to thoroughly compare two LES models for Martian Planetary Boundary Layer convection. We corrected the paper accordingly to clearly reflect this modest goal and state the remaining goals. We changed the title of the paper. We also added an entire

section "Challenges of LES intercomparisons and suggestions for future studies" where we summarize the difficulties and ideas mentioned in the paper and in the reviews. We strongly believe the revised version will be a useful milestone for the community of Martian modelers, if not a definitive reference on Martian model intercomparisons, which is clearly an ambitious goal for a future study funded much more extensively than the preliminary one we performed. We add, attached as a comment, a pdf document comparing the previous and the new version of the manuscript (in order for the editors and reviewer to better track the changes). Note that the references are not displayed in this pdf document (made with latexdiff which does not take references into account and also get confused sometimes with the order of the sections) but of course, in the submitted paper, references and section ordering are fine.

**1) The horizontal size of the modeling domain used in this study (a fatal problem): When performing LES studies of the convective boundary layer (CBL), it is imperative that the horizontal size of the domain is large enough so the periodic boundary conditions cannot influence/contaminate the solution computed for the domain interior. The authors suggest that they have followed the guidance of Mason (1989) to achieve this, but they haven't. For typical square LES domains, a generalized "rule of thumb" (as was used, if not clearly stated, by Mason (1989)) is to design the grid so the length of the domain side is 3x the size of the largest eddy that will be resolved by the simulation. For an afternoon Mars EDL related investigation, the size of the largest eddy scales as the maximum depth of the CBL, which can reach 10 km (thus, 30 km would be an appropriate lateral size). It's an unfortunate reality of LES for Mars that, to sufficiently resolve the range of smaller eddies and the complexity of convective structures (with the underlying desire being to capture the energy spectrum), a grid-spacing of 50 m (as was used here) is needed. For a square LES domain with a 50 m gridspacing, the number of computational locations would then be 600x600. In this work, with only 145x145 computational locations, the lateral size of the domain is just 7.25 km (this is far too small), and the problems that will be**

**created in this approach will contaminate the analysis and intercomparisons that could be performed at a time of day with a deep CBL. For Mars, especially once any wind profile is introduced to force the simulation, this 3x aspect ratio rule likely should be considered a minimum, and this is due to dramatic variations seen across the diurnal cycle of the scale of convective structures as a function of time and height in the domain. This poses a real problem for LES of the Martian atmosphere, and any study desiring a high-quality LES model intercomparison should err on the side of having completely eliminated the possibility of a contaminated simulation due to a very deep CBL that will interact with the periodic boundary conditions. Most certainly, the top of the modeling domain should also be well above the top of the CBL. As used here, a top of 12 km is probably sufficient, although a few km higher would be preferred. For the goals of this manuscript, the 145x145 number of computational locations, with a grid spacing of 50 m, is a fatal problem. It's easy to see that LES studies involving the CBL on the planet Mars almost certainly require the use of massively parallel architectures.**

We thank the reviewer for pointing out this issue. Our choice of domain size is a tradeoff: we want the domain to be as large as possible to represent correctly the growth of the convective PBL, but we need it to run in a reasonable time and with a reasonable size of outputs to be able to carry out the intercomparison and sensitivity study. The 145x145 setting also corresponds, as imperfect as it is, a common setting employed in existing LES studies for Mars.

We slightly disagree with the reviewer on this "domain extent" issue being a fatal problem for our study. We based our LES configuration on the modeling experience reported in Spiga et al. QJRMS 2010, which shows that reasonable estimates of PBL growth, convective plumes, eddy heat flux, can be obtained with the configuration adopted (compared to LES performed with wider domains).

In addition to this, we showed in Figure 6 and 7 horizontal sections at 11:00 local time

where the PBL depth is about 2 km, which is reasonably close to the Mason 1989 rule of thumb. In those Figures, it can be observed that at least two (maybe three) convective cells are enclosed across one dimension of the LES domain.

We agree, however, with the reviewer that the domain extent remains a serious matter to be considered when performing LES. We would certainly not state that our LES setting is the perfect setting to be followed by future studies. We thus followed the reviewer's advice and we performed a LMD LES simulation with a raised model top (keeping vertical resolution similar as in our reference setting) and a much larger domain extent (250x250, progressing towards more grid points would involve computing resources that are not easily available at the time of writing). Results at local time 11:00 remained unchanged, but 25

Thus, we are convinced that our results are still valid and will be useful to the community. Still we do not want to downplay the crucial comment made by the reviewer here. We thus changed the text to better describe our choice of domain and the tradeoff, and we discuss this important comment of the reviewer in the sensitivity study (using the above-mentioned additional 250x250 LES) and in the new section Challenges.

**2) dust used during the intercomparison phase (thoughts/suggestion): As described by the authors, without managing the issue of dust and its differing treatments between the two models, an intercomparison of LES models could reduce itself to an 'intercomparison of radiation schemes', not of the dynamical cores as is desired (this is also true for mesoscale and global climate model intercomparisons). The authors did put effort into getting the LMD model to show an improved agreement with the SwRI model to facilitate their intercomparison, although I wasn't fully convinced this effort was actually successful. From a perspective of do it as simply as possible, it's unclear why the authors didn't just run/compare the two models (the primary focus of this manuscript) with no dust loading at all, a dust-free atmosphere. Even with dust properties modified so ground temperatures come into far better agreement, there was no**

**discussion about heating rate profiles, that they had also come into better agree-
ment as a result. It is much easier (and more straightforward) to use a dust-free
atmosphere for the intercomparison focus of this effort, eliminating the compli-
cation of the radiative properties of dust in the atmosphere, and the probable
non-linear response to this change the the heating rates as a function of height.
For the secondary aspect of this effort (the prediction of the EDL environment
for the Sciaparelli spacecraft), dust would be reintroduced in both models, and
LES results would need to be compared with mesoscale model results (presum-
ably form both of the parent models). Most certainly, a primary reason to use
LES is to both improve and qualify our confidence in (and understanding of) the
results from mesoscale models, specifically the performance of PBL schemes
being used (since there is no PBL scheme in LES). Moreover, and I believe this
is important, the use of mesoscale model results would allow the ability to char-
acterize the larger-scale environment in which the LES was being performed. It's
important because the larger-scale regional/local circulation can dramatically af-
fect the evolution of the CBL. Careful consideration of this for site-specific LES
is not addressed by any authors to date, whereas local strong subsidence has
been shown by Tyler and Barnes (2015) to be very important to the development
of the CBL, with actual evidence of this (Moores et al., 2015). How can this real-
ity be incorporated into LES, and is it even possible? For Mars, LES has unique
challenges, and a manuscript sufficiently well thought-out, that addresses some
of these issues head-on with some new approaches would be most-welcomed
and likely important to the community.**

About performing dust-free atmosphere: we agree with the reviewer that this can be
a simpler method to investigate the differences between the LMD and SwRI LES dy-
namical cores (respectively the WRF and RAMS terrestrial models). Our comparison
study is, however, intended to provide an assessment of the spread of results of Mar-
tian LES as realistic as possible to be used for EDL studies. This makes the dust-free
LES difficult to justify in this context, where realistic temperature profiles accounting for

the radiative impact of dust (which is strong on Mars) are needed.

About performing LES with mesoscale models: we agree with the reviewer that this shall be an overarching objective for future LES studies. Our goal here is, nevertheless, to assess differences between existing LES. To the extent of our knowledge, there is only one LES study which accounts for evolving background conditions following a diurnal cycle [Tyler et al. 2008]; and no published Martian LES study features the use of predictions from mesoscale modeling to represent time-evolving conditions for LES. Instead of trying innovative LES settings, we based our comparisons on the usual settings adopted in most LES studies thus far. We certainly agree, though, with the reviewer that LES studies in the future shall try and incorporate a better account for the diurnal variability in the regional and global dimension, which is expected to be strong on Mars.

We tried to revise our manuscript accordingly, to reflect those perspectives and challenges, although we unfortunately cannot change our paper to make it as groundbreaking as what the Reviewer is suggesting. The scope of our study is perhaps less ambitious than the community is hoping for; yet we still believe we made a significant first step that is worth being reported to inspire better studies of this kind in the future.

We added a discussion about this point in our new section "Challenges", based on this comment provided by the reviewer.

**3) subgrid mixing parameterizations (thoughts/suggestion): Upon looking at the results provided, it's easy to agree with the authors that subgrid mixing in the LMD model is much stronger than it is in the SwRI model. A short section in the manuscript suggested the authors did experiment with the subgrid mixing strength in the LMD model. Unfortunately no results were shown to indicate the degree of change seen towards what I would expect to be a much more 'noisy' solution (more like that in the SwRI model). Were such changes seen in that exercise, and if not did the authors try to completely disable the subgrid mixing**

**to insure it had indeed been modified?**

We tested the model sensitivity to the subgrid mixing coefficient but it has little effect on the results. By decreasing the coefficient by a factor of 1000, we obtained a slightly more noisy solution and but no change of PBL height. Decreasing more the coefficient does not affect the results more than that. Actually, it is not trivial to completely disable the subgrid-scale mixing in the LMD LES based on WRF, because the WRF dynamical scheme possesses a discretization term which is naturally diffusive. It would certainly help to disable this term completely, but this remains a difficult task.

We detailed more this point in section 5.1 and discuss it in section Challenges. All in all, we would like to show in our paper that even when the radiative transfer is matched between two Martian LES models, the predictions of both models can be remarkably different for reasons related to the choice of dynamical core itself.

**Analogously, subgrid mixing schemes are to LES as PBL schemes are to mesoscale models; and, both are fundamentally untested in regard to being used in atmospheric modeling for Mars (designed for and tested in terrestrial modeling). I believe we should have a bit more trust in the subgrid mixing scheme of LES (as being fundamental) than the PBL scheme of a mesoscale model, although doubt must still be raised in regard to the isotropic nature of vertical mixing compared to horizontal in a deep and energetic CBL. How do individual schemes address this? The results here serve to elucidate the central importance of subgrid mixing schemes in LES model intercomparisons, where a thorough intercomparison would investigate a wide range of strengths of subgrid mixing. We should be able to show that results become 'smooth' on one hand (as in the LMD results) and very 'noisy' on the other hand (as in the SwRI results) with variations in the strength of this mixing. The idea would be to investigate the point at which the energy spectrum became sufficiently corrupted and/or modified. Possibly, a balancing of the subgrid mixing schemes between the two models, in a nodust scenario such that the energy spectra became more**

**similar, would be a logical framework from which to begin this intercomparison. This has never even been tried by investigators, and as so clear in the results presented, it's a central question that was not sufficiently addressed.**

We admit that the reviewer is, again, perfectly right. This is a difficult question to address, though (that could easily fill an entire paper). We might be wrong, but we thought such detailed analysis of the energy spectrum simulated by both dynamical cores would take our paper too far from its initial goal which is a preliminary step towards a comparison of Martian mesoscale and LES models – a topic seldom, if not never, discussed in the literature. Reading this comment makes us realize that our initial submission may have seemed too ambitious in its intent, whereas our goal is to offer a preliminary LES comparison to the community, with a clear identification of the remaining challenges to perform a truly complete intercomparison of Martian models. These comments have thus been mentioned in our section "Challenges".

**MINOR THOUGHTS/ISSUES:**

**The vertical grid of the SwRI model is actually quite different, with heights fixed above ground in a sigma z (not sigma p) formulation, quite different form the vertical grid used in the LMD model. It actually leads to a "pressure cooker" effect in MRAMS. For comparison of these two models, it is worth acknowledging that there shouldn't be any surface pressure variation in the LMD model, while there should be in the SwRI model.**

It is indeed the case, although the pressure variations in the SwRI model are relatively small. This comment has been mentioned in our section "Challenges".

**Without a doubt, spacecraft payloads are only going to get more massive, which renders the sensitivity of any spacecraft to atmospheric turbulence quantified during the EDL phase by LES in the CBL (such as for Phoenix or Insight) a moot point. The LES modeling that was done for Phoenix was driven by the possibility of oscillations in the parachute/lander system that (according to my understand-**

**ing) are not problematic for either the MSL or the Mars 2020 spacecraft. I'm unaware of any LES modeling being requested/performed for Insight. In the near future, SpaceX and others will be landing massive payloads; and, the EDL trajectories for such payloads will cause the spatial variation of the CBL depth (as modified by topographic circulations) to be far more important during the final phase of 'flight' (quite low for large horizontal distances to make the greatest use of larger air densities). Results from LES are likely to provide the turbulence spectra that turns out to be crucial for autonomous navigation software, etc: : : My sense at this point is that the reason for high-resolution LES modeling may have much more to do with autonomous robotic flying exploration vehicles, such as Mars Helicopter: http://www.jpl.nasa.gov/news/news.php?feature=4457**

This is a very good point, especially since our paper discusses LES modeling that was performed in the context of an EDL analysis. We added in the revised text a few sentences about this discussion.

**The manuscript was not clear on whether the surface layer scheme truly only acts on the lowest model layer acting with the ground for turbulence closure. I know this is the case for the OSU Mars LES model and believe it must also be so for the LMD and SwRI LES model. It's a point worth making, especially since the authors see differences in the near surface wind and/or temperature profiles. The structure near the surface is quite possibly another means through which the subgrid mixing can be examined in an intercomparison.**

Indeed, what the Reviewer describes is a standard way to account for surface layers in our model too. We completely agree that surface layer is another plausible source for the observed discrepancies between the LMD and SwRI LES models. We mention this possibility in our new section "Challenges".

**The authors certainly recognize that systematic intercomparisons between mesoscale and/or LES models have not been carried out, possibly because no**

**funding entity seems to think that such intercomparisons are important. I acknowledge and appreciate that this work has tried to remedy this situation. Since atmospheric dynamics are actually so much more important on Mars in relation to physical processes than they are on the Earth, raising awareness here is important. Possibly, we are stuck in a terrestrial modeling paradigm that we need to escape to improve our mesoscale and LES modeling of the Martian atmosphere.**

This is a comment that is impossible to include in a scientific paper, but we fully agree with the reviewer. This is exactly what we intend to do with our paper (and this is why we submit a revised version reformulated towards identifying all remaining challenges): provide a starting point to further intercomparison studies to try to remedy a situation (even a paradox shall we say) where mesoscale modeling are being extensively used to assess the conditions for EDL landing, yet 1) observations to validate those models are still scarce and 2) it is difficult to find funding, time, and human resources to make the necessary intercomparisons studies for such important tools. This is why we believe our study, albeit admittedly imperfect, is a necessary first step which is essential to publish.

**Finally, even though it only leads to greater computational burden, it is definitely worth considering the value of running the LES model through one full diurnal cycle before examining results from the second full diurnal cycle. Such 'spin-up' does affect the results, with the second diurnal period quite different. When background forcing is added, the second diurnal cycle can be quite different from the first, where the morning air temperature profiles (before the onset of convection) differ significantly, which noticeably affects the growth/evolution of the CBL. In the diurnal cycle, there is another weakness of LES (the highly stable regime, even more stable in the atmosphere of Mars) to investigate and better understand. We strive for the most realistic/honest results from our models; and, in this, intercomparison is an art form that I hope efforts such as this convince us to practice more enthusiastically.**

Addressing this question is beyond the scope of our paper in which we only focus on the daytime convective boundary layer, because this is the focus of the vast majority of the published Mars LES studies. There is actually another reason: simulating the small-scale turbulence in highly stable conditions is still challenging also on the Earth.

The reviewer makes an interesting point though: differences may be found when comparing the second LES day with the first LES day. We left this discussion apart because we felt it has already been published elsewhere [cf. Tyler et al. 2008], and we would not have been able to provide new elements on this question.

Please also note the supplement to this comment:
http://www.geosci-model-dev-discuss.net/gmd-2016-241/gmd-2016-241-AC2-supplement.pdf

**Supplement:**

**A comparison of Large-Eddy Simulations of the Martian daytime convective boundary layer: sensitivity study and challenges**

Tanguy Bertrand[1], Aymeric Spiga[1], Scot Rafkin[2], Arnaud Colaitis[1], François Forget[1], and Ehouarn Millour[1]

[1]Laboratoire de Météorologie Dynamique (UPMC/CNRS), Paris, France
[2]Southwest Research Institute, Boulder, CO, USA

*Correspondence to:* Tanguy Bertrand (tanguy.bertrand -a- lmd.jussieu.fr)

[revised manuscript text omitted]

A systematic intercomparison between  all existing Martian LES models is still yet to be carried out to further characterize those differences.  Such an effort to compare all available models is beyond the scope (in term of human, time and funding resources) of the present study  (see section **??**). Here we propose, as a necessary first step

55 towards a true intercomparison, an unprecedented systematic comparison between two existing Martian LES models based on two distinct hydrodynamical solvers.

The need to evaluate the differences predicted by two distinct Martian LES is threefold:

1. Contrary to observational data, an estimate of the uncertainties of LES  diagnostics is still lacking. This is especially critical for the studies of atmospheric hazards during EDL, given the central role that LES  diagnostics play in those studies and the relative paucity of available data to characterize the Martian PBL dynamics (e.g. for vertical winds). Comparing LES models would provide guidance on the range of model variance (i.e. the spread in modeling results for a similar Martian site and season), thereby enabling an optimal EDL design for both landing spacecraft and definition of landing ellipse.

2. Carrying out a LES  comparison would highlight discrepancies between results and help to identify the specific areas in which model improvements would be the most helpful. This overarching goal is beneficial for the whole Martian science. For instance, turbulent wind variability (i.e. "gustiness") plays an important role in controlling dust lifting on Mars (e.g., **?**, and references therein). Since turbulent wind measurements on Mars are very incomplete, LES predictions are still being an important source to assess the wind conditions on Mars associated with dust lifting (**?**). More generally, the continued development of Martian LES models is also of crucial importance to better understand the mechanisms responsible for heat and momentum transfer both by daytime PBL mixing and surface-atmosphere interactions. Following the tendency drawn by terrestrial studies, Martian LES predictions are more and more used to build and improve PBL parameterizations in Global Climate Models (GCMs) for Mars (**?**).

3. Since Martian LES rely on hydrodynamical solvers inherited from terrestrial studies, confronting those models to the intense PBL convection on Mars (compared to the Earth, cf. **?**) provides a stringent test for those solvers.  A comparison study of two or ideally more Martian LES will ultimately be a strong driver of improvement for the atmospheric models used to carry out LES on Earth, and in an increasingly diverse range of planetary conditions (e.g. Venus LES, **???**).

In this paper, we compare the LES results obtained by, on the one hand, the Laboratoire de Météorologie Dynamique (LMD) Martian mesoscale model (**??**) and, on the other hand, the  Southwest Research Institute (SwRI) Martian mesoscale model (**??**). We develop a strategy which makes our  comparison study the first one of its kind for Mars: we ensure that similar physical constants and radiative forcing are employed in both models before performing a comparative analysis of LES results and conclude on the performance of the two dynamical solvers in predicting Mars' PBL convective motions. We further complement this  comparison study by an exploration of the sensitivity of the convective PBL predicted by the LMD LES to surface thermophysical properties (e.g. albedo), ambient wind, and atmospheric dust loading.

 The present LES comparison has been performed in the context of the European Space Agency (ESA) ExoMars 2016 mission (hereinafter referred as ExoMars) with the aim of providing constraints for the EDL of the ExoMars Demonstrator Module (EDM, also named Schiaparelli). LES modeling  has therefore been performed at the ExoMars landing site, namely in the Terra Meridiani region (latitude $-1.82°$N, longitude $-6.15°$E),  at the landing date in northern autumn (solar longitude $L_s = 244°$).

In  section **??**, we describe the LMD and SwRI models used in this LES comparison. In section **??**, we provide details on the  comparison strategy, and how we reached similar radiative forcing both in SwRI and LMD LES. The results of both the Martian LES  comparison and sensitivity study are discussed in section **??** and **??** respectively. Finally, in section **??**, we discuss the challenges of LES intercomparison and we suggest a possible path forward for future LES studies.

**2 Models description**

We provide here the key points to describe the two models used for our LES comparison. Further details about each model can be found in the references provided in this section.

Both LMD and SwRI Martian LES models have been built independently by adapting terrestrial mesoscale models to the Martian case, with the coupling of specific physical models (namely, radiative transfer and soil model) initially developed for Martian GCMs:

- LMD LES are performed using the LMD Martian Mesoscale Model (**??**), based on the Weather Research and Forecast (WRF) model and its fully compressible non-hydrostatic dynamical core (**??**), combined with the comprehensive set of physical parametrizations of the LMD GCM (**?**).

- SwRI LES are performed using the Mars Regional Atmospheric Modeling System (MRAMS), a nonhydrostatic Martian mesoscale model developed at SwRI (**??**) and based on the terrestrial RAMS dynamical core (**?**), in which physical parameterizations are inherited from the Martian NASA Ames GCM (**?**).

The two LES models not only use very distinct different radiative transfer and soil  models (inherited from GCM), but also use different dust scattering properties (at the time the runs for this study were carried out, **?** for SwRI LES vs. **?** for LMD LES), which can lead to significant departures in the predictions of atmospheric temperatures.

[revised manuscript text omitted]

SwRI models. We chose to vary 1. the dust extinction efficiency $Q_{ext}$ (or "thermal infrared opacity", at the reference infrared wavelength) and 2. the visible single scattering albedo $w_0$, which quantifies dust "brightness" (further details and explanations in **??**). For the sake of simplicity, we define the ratios $S$ and $C$ of the corrected value over the reference value for respectively $w_0$ and $Q_{ext}$ in the LMD model. The higher $S$, the brighter dust; the higher $C$, the stronger absorption by dust in the infrared.

**??** shows the sensitivity of the temperature profiles to $S$ and $C$. Here, the comparison of profiles is focused above the PBL because in 1D, the PBL is parameterized while it is resolved in LES (see next section for comparisons within the PBL). We performed a comparative study for all local times, given that LES runs span stability conditions from highly unstable around noon to highly stable in early morning and late afternoon. For the sake of illustration, we show in the aforementioned figures typical nighttime and daytime temperature profiles at local times $0:00$ and $12:00$. In the clear atmosphere case we adopted (setting dust opacity to $0.2$), dust brightness $S$ has a strong impact on the temperature profiles: a decrease of $20$ K is obtained in nearly the whole atmospheric column when increasing $S$ from $0.92$ to $1.04$. Indeed bright dust means higher dust albedo, hence increased reflection of solar flux to space, and colder atmosphere and surface. The dust extinction ratio $C$ has a lesser impact (though not negligible) than $S$ on the thermal structure, as shown by **??**. This is expected for a clear atmosphere, where dust absorption in the infrared is not significant enough to cause a strong heating of the atmosphere (contrary to dust storm conditions). This sensitivity study shows that the values of the dust visible scattering albedo and extinction efficiency should be decreased to simulate the effect of dust aerosols both darker and more strongly absorbing in the infrared. The effect of this correction is to warm the atmosphere (both during daytime and nighttime).

As the tuning of these dust parameters does not change significantly the surface temperatures (because of the relatively clear atmosphere considered), the surface thermal inertia has also been changed in the LMD model so that it replicates the SwRI diurnal cycle, at least during daytime when the surface temperatures impact the PBL. Many optimization loops have been performed to replicate in an optimal way the  SwRI temperatures at all local times, taking into account

[Figure]

**Figure 4.** Same as **??** with $S = 94\%$ and $C = 70\%$ in the LMD 1D model.

the change of surface temperature as well.  Values of $S = 94\%$ and $C = 70\%$ are used, in order to bridge the $5 - 10$ K gap between the LMD and SwRI temperature profiles ( **??**). In addition, by increasing the thermal inertia up to 300 tiu (which remains close to the surface conditions encountered at ExoMars landing site), we decrease the LMD surface temperatures  by about 8 K around noon and replicate the SwRI daytime surface temperatures, as shown in  section **??** by **??**.

However, there is still a 20 K gap in the daytime near-surface temperatures that cannot be reduced (over a range of plausible parameters) by optimizing the radiative forcing as it is the case for the rest of the PBL. This suggests that differences in the formulation of surface layer schemes in the two models (see (**?**) for MRAMS and (**?**) for LMD LES) might explain such a difference. This remains to be explored, as discussed in section **??**.

**4  Comparison of LES results**

This section compares the LMD and SwRI LES results, obtained using the settings described by Table **??** and Table **??**, with the modified radiative transfer properties detailed in section **??** which allow a similar radiative forcing between the two models ($S = 94\%$, $C = 70\%$ and thermal inertia = 300 tiu). Two main test cases are considered: LES devoid of any ambient wind, and LES with an ambient wind of 15 m.s$^{-1}$, thereby providing typical LES results of Martian daytime boundary convection in windless and windy conditions at the ExoMars landing site.

**4.1  Forcing of PBL activity**

The PBL activity and its evolution are controlled by the surface temperature and the surface incident solar flux. As described in section **??**, the preliminary steps of convergence of both models enabled a similar forcing to be reached, as shown by **??**. During daytime, discrepancies between LMD and SwRI LES surface temperatures are less than 1 K, and we checked that the shortwave and longwave incident solar flux are similar at all local times. In addition, we  checked that the atmosphere above the PBL

265 is controlled by similar radiative forcing in both models, as is found in the previous section with the 1D models. With these settings, any difference observed in the LES results (PBL depth, maximum updrafts...)  should not be related to differences in "background" radiative thermal structure, but rather to different dynamical approaches in both models.  Other possible sources of differences are discussed in section ??. For example, as mentioned above, the two models use different PBL  surface layer schemes, which  can impact the results (see  sections ?? and

270 ??).

**4.2 Comparison of typical LES wind structures**

In this section we analyze the turbulent wind structures in order to verify if they are similar in both models and consistent with typical LES diagnostics. Note that the ambient wind is westward in the simulations.

**4.2.1 Wind structures and small scale variability**

275 ?? shows horizontal sections of vertical winds at about 250 m and 2 km above the surface and at local time 11:00, obtained without ambient wind. Both models predict a horizontal organization of vertical winds into polygonal cells with narrow updrafts on the ridges of the cells and large subsidence in the middle of the cells, with same spatial dimensions and structure sizes. These are a typical pattern of the PBL which remain in conformity with typical LES studies. At 250 m altitude, maximum updrafts and downdrafts reach up to 8 m.s$^{-1}$ and -6 m.s$^{-1}$ in both models, with slightly higher values over few grid points in SwRI

280 results. At 2 km altitude, where updrafts are more vigorous, the differences between both results are more pronounced. LMD LES  predicts a maximum updraft value of 8 m.s$^{-1}$ while SwRI LES results show values twice higher, around 15 m.s$^{-1}$ (see  section ??). Because of mass conservation, downdrafts are also found to be more vigorous in the SwRI LES than in the LMD LES.

?? and ?? show horizontal sections of horizontal wind amplitudes at local time 11:00, at about 5 and 250 m above the
285 surface, in the case of an ambient wind of 0 and 15 m.s$^{-1}$ respectively.

In the case without ambient wind, horizontal wind structures are similar in both models and typically show gusts in all directions related to the convective cells in the PBL. However, it can be noticed that the variability at very small scale (i.e. a few grid points apart) is much higher in SwRI results than in LMD results. This higher small-scale variability appears in all SwRI simulations . They may be related to the quantitative discrepancies  seen in PBL diagnostics
290 (section ??) and summarized in Tables ??-??. This is further discussed in section ??. Here, maximum horizontal wind amplitudes at 5 m above surface are found to be in the range 6-8 m.s$^{-1}$ in both LMD and SwRI LES, although SwRI results also show maximum values at 12 m.s$^{-1}$ over a few grid points. At 250 m above surface, horizontal winds reach 6 m.s$^{-1}$ in both models (9 m.s$^{-1}$ over a few grid points in the SwRI model).

When a sufficiently strong ambient wind is added (typically more than 10 m.s$^{-1}$), it distorts the organization in polygonal
295 cells and forces the gusts to propagate in the same direction, as shown by ??. In this case, LMD wind amplitudes are found to be larger than SwRI values, especially 250 m above the surface, although the wind structures remain similar in both LES. Maximum values for horizontal wind are 16-18 m.s$^{-1}$ in both LES.

This difference in altitude is highlighted by **??**, showing the mean profile of horizontal winds in both models between 11:00 and 12: 00, in the simulation with 15 m.s$^{-1}$ ambient wind. SwRI results show weaker horizontal winds than LMD results in
300   the whole PBL. Although these differences could be related to the different intensity of the convection, they may also be related to the different scheme of near-surface stability. The large differences encountered in the first hundreds of meters reinforce this possibility  (see section **??**). It can also be noted that SwRI results show a deeper mixed layer, which is consistent with the overall weaker winds (momentum is mixed through a deeper layer).

**4.3 Comparisons of PBL diagnostics**

305   LES results compared in this section are the PBL depth, the turbulent kinetic energy (TKE) and the maximum updrafts and downdrafts within the PBL, which are key characteristics of the convection inside the Martian PBL. The PBL depth is directly and positively correlated to the intensity of its dynamics while the TKE is a measure of turbulence intensity and is directly related to the transport of heat and momentum through the PBL. An increase of TKE denotes a more turbulent PBL. Updrafts and downdrafts also enable  the comparison of the intensity of the PBL turbulence in both models. Table **??** and
310   Table **??** summarize the results.

**4.3.1 Planetary boundary layer height**

**??** shows that the daytime PBL depth grows faster and reaches a higher vertical extent in the SwRI LES than in the LMD LES. By the end of the afternoon, the PBL depth reaches 7.5 km (SwRI) and 4.8 km (LMD) without ambient wind and slightly more in windy conditions. These results confirm that the convection in the SwRI LES is more vigorous. In fact, the
315   maximum PBL depth obtained in the SwRI LES is close to the highest values that could be found on Mars and measured through  radio occultation, which is about 8-10 km (**?**). It should be pointed out, however, that these simulations contain none of the realistic physics that would tend to suppress a PBL, notably large-scale subsidence or regional circulations (e.g. as experienced in Gale Crater, see **?**). The turbulent convection is active until the end of the afternoon (typically 16:30-17:00) and suddenly stops (the PBL collapses) when the surface becomes colder than the atmosphere above it
320   (nighttime inversion). For some of the simulations, this occurs before 17:00, explaining the sharp drop in **??**.

**4.3.2 Turbulent kinetic energy**

The differences in the vertical extent of PBL mixing between both LES can also be inferred from the variations of (resolved) TKE, as shown by **??**. Although the convective activity rapidly declines at same local time 17:00 and the maximum TKE values occur around same local times 13:00-14:00 in both models, larger TKE values are found in the SwRI LES, in consistency
325   with previous results,. Without ambient wind, the maximum TKE in the PBL reaches 12 m$^2$.s$^{-2}$ around 1.5 km altitude in LMD results, while it reaches about 21 m$^2$.s$^{-2}$ around 3.5 km altitude in SwRI results, a factor of two more intense. Similar differences are found in the LES with windy conditions.

| LES without ambient wind | LMD | SwRI | Difference (%) |
|---|---|---|---|
| PBL  height (km) | 4.8 | 7.5 | 56 |
| Turbulent  heat flux (K.m.s$^{-1}$) | < 1.25 | < 3.2 | 156 |
| Turbulent  kinetic energy (m$^2$.s$^{-2}$) | < 12 | < 21 | 75 |
| Maximum  updraft speed (m.s$^{-1}$) | < 15 | < 24 | 60 |
| Maximum  downdraft speed (m.s$^{-1}$) | < 8 | < 12 | 50 |

**Table 3.** Maximum PBL diagnostics values from the LMD and SwRI LES without ambient wind

**4.3.3 Maximum speed for convective updrafts**

In line with previous results, **??** shows that maximum updraft speeds obtained from SwRI LES are higher than those from LMD LES, with a ratio of about 1.5-1.8, all over the mixing layer depth between local times 11:00 and 17:00. In LES without ambient wind, maximum speeds reach 13-15 m.s$^{-1}$ in LMD results around local times 12:00 – 14:00 between altitudes 2 to 4 km while they reach 20-24 m.s$^{-1}$ in SwRI results between altitudes 2 to 6 km at same local times. In windy conditions, maximum updraft speeds in both LMD and SwRI LES remain in similar ranges. This is expected given that SwRI has finer structure (narrower updrafts, presumably less diffusive).

**4.3.4 Maximum speed for convective downdrafts**

In both LES, maximum updraft speeds (**??**) are in comparison larger by a factor 2 than maximum downdraft speeds (**??**). This is a consequence of the organization of turbulence in cells with narrow updrafts and broader downdrafts. Both LES predict maximum downdraft between local times 13:00 and 15:00. Without ambient wind, downdraft up to 8 m.s$^{-1}$ are predicted by the LMD LES from 200 m to 5 km above the surface while the SwRI LES predicts values up to 12 m.s$^{-1}$. In the LES with ambient wind, these values slightly increase to 9 m.s$^{-1}$ and 15 m.s$^{-1}$ respectively.

**4.3.5 Distributions of vertical wind speeds**

**??** and **??** show the distribution of vertical wind speeds obtained between local times 13:00-14:00 and altitudes 250-5000 m, for both LES without and with ambient wind respectively. In the windless case, 95% of vertical wind speeds are in the [-6,6] m.s$^{-1}$ range for LMD and [-9,9] m.s$^{-1}$ for SwRI. In the windy case, 95% of vertical wind speeds are in the [-7,7] m.s$^{-1}$ range for LMD, and [-11,11] m.s$^{-1}$ for SwRI. Therefore, the strongest vertical winds represent a very low probability. As an example, in the SwRI LES, both the maximum downdraft value of 11 m.s$^{-1}$ and the maximum updraft value of 28 m.s$^{-1}$ represent less than 0.01% of all values.

| LES with 15 m.s$^{-1}$ ambient wind | LMD | SwRI | Difference (%) |
|---|---|---|---|
| PBL  height (km) | 5.3 | 8.1 | 53 |
| Turbulent  heat flux (K.m.s$^{-1}$) | < 1.7 | < 3.4 | 100 |
| Turbulent  kinetic energy (m$^2$.s$^{-2}$) | < 15 | < 34 | 126 |
| Maximum  updraft speed (m.s$^{-1}$) | < 15 | < 22 | 46 |
| Maximum  downdraft speed (m.s$^{-1}$) | < 9 | < 15 | 60 |

**Table 4.** Maximum PBL diagnostics values from the LMD and SwRI LES with 15 m.s$^{-1}$ ambient wind

**5 Sensitivity simulations**

This section presents a sensitivity study of the Martian daytime PBL properties in the LMD LES model to dust loading, surface
albedo, ambient wind and subgrid scale diffusion coefficient, using the same reference settings as described in  section
**??**, without ambient wind. This enables to  explore the role of forcing  of the Martian
PBL  and to assess if an uncertainty of
one parameter can explain the discrepancies evidenced between the LMD and MRAMS LES results. Furthermore,
 such a sensitivity study has seldom been detailed in the existing literature.

**5.1**

**5.1 Sensitivity to dust opacity**

LES have been performed using distinct dust optical depths of 0.2 (reference), 0.6, 1 and 3. Comparing LES predictions in those
three cases reveals dramatic differences in the strength of the boundary layer convection. **??** shows how the daytime evolution
of boundary layer depth is influenced by dustiness in the Martian atmosphere. Firstly, the mixing layer is of significantly lower
vertical extent for higher dust opacities (which is true during the whole day). At local time 14:00, boundary layer depths are
respectively 3.8, 3, 2.5, 1.7 km for dust loadings 0.2, 0.6, 1, 3. This behaviour originates from dust absorption of solar radiation
in the visible leading to decrease in surface temperature. Moreover, stability in the lowest layers of the atmosphere is enhanced
by dust heating, which leads to a less vigorous boundary layer turbulent convection.

Secondly, the PBL is collapsing earlier in the afternoon when dust opacity increases: while the PBL depth is still maximum at local time 17:00 in a clear atmosphere, it starts to slightly decay for a dust opacity at 0.6 and more distinctively decrease for an opacity at 1 (the maximum vertical extent in this case is attained around local time 15:30). The most extreme case (dust opacity at 3) shows a very limited growth of the convective boundary layer, which peaks at the very low value (with respect to Martian standards) of 1.7 km between local times 13:00 and 14:00, before a rapid collapse of the turbulent convection occurs at local time 14:00-14:30.

Both the limitation of convective activity and the displacement of its maximum towards earlier afternoon associated with an increase in dust optical depth can also be assessed by the comparison of maximum (resolved) TKE shown on **??**. Differences in TKE between the clear and extremely dusty cases are about one order-of-magnitude – even differences between the opacity 0.2 and 1 are significant (50 to 60% decrease). It is interesting to note that, in theory, dust radiative heating by absorption in the visible (plus a smaller contribution in the infrared) should cause TKE and updraft speeds to increase with dust opacity. Present LES results with complete radiative transfer show this effect does not significantly compensate the aforementioned influence of dust loading on surface temperature and atmospheric stability. Overall, the dustiness of the Martian atmosphere strongly determines the strength of boundary layer convection and this has important consequences on conditions for EDL systems as ExoMars landing between 14:00 and 16:00. This is exemplified by maximum vertical wind speeds (not shown). Maximum updraft and downdraft values throughout the whole day vary dramatically with dust opacity: updrafts of 15, 12, 10, 4 m.s$^{-1}$ and downdrafts of -9, -7, -5, -2 m.s$^{-1}$ are predicted by LMD LES for dust opacities 0.2, 0.6, 1 and 3. Again, convection for an opacity at 3 is severely limited. Vertical winds are lower at 16:00 than at 14:00 in all cases, but the differences are more prominent between the two local times when more dust is suspended in the Martian atmosphere.

Finally, it is important to note that if the dust opacity values follow a geometric progression which implies a linear increase in atmospheric heating, variations of PBL depth, TKE and maximum winds with dust opacity are not linear. The sensitivity of PBL turbulence to dust opacity is a complex combination of dust influencing surface temperature, atmospheric stability and maximum turbulent heat flux as well as turbulent motions adjusting to those various modified forcings. Only LES employing full radiative transfer can address this complexity.

**5.2   Sensitivity to surface albedo**

In order to assess the sensitivity of the PBL to surface conditions, the LMD LES has been tested with surface albedo value of 0.1, 0.21 (reference), 0.4 and 0.6. A value of 0.1 is extreme, although not unrealistic, and causes surface temperatures to be significantly warmer, hence boundary layer convection to be more vigorous, as shown by **??** and **??**. A lower albedo not only  causes the sensible heat flux to be larger, but also the infrared radiative flux emitted by the surface and absorbed by $CO_2$ and dust in the lowermost atmospheric layers to be larger. The PBL depth predicted by LES is about 20% higher for the 0.1 albedo case, and 35% lower for the 0.4 case, compared to the 0.21 case. Similar conclusions apply for the maximum TKE predicted. The PBL convection on a more reflective surface is very limited, with a height of 1.5 km for an albedo up to 0.6.

**5.3 Sensitivity to ambient wind**

The sensitivity of the PBL to the ambient wind is addressed by running the LMD LES with 0, 15 (both reference runs) and 25 m.s$^{-1}$ ambient wind. In the afternoon, the wind enhances convection by increasing the surface-atmosphere heat and momentum transfer in the surface layer. Larger wind speeds thus yield higher values for turbulent heat flux, and higher values of TKE (according to the TKE equation). **??** shows that the PBL in windy conditions is especially vigorous between local times 12:00 and 15:30. In addition, the boundary layer convection appears to start earlier in the morning in windy conditions. The maximum TKE predicted by LES in the afternoon is about 20% higher for the 15 m.s$^{-1}$ case (40% for the 25 m.s$^{-1}$ case) compared to the no-wind case. In such conditions, the maximum PBL depth is higher by about 500 m (10%) in the 15 m.s$^{-1}$ case and 700 m (15%) in the 25 m.s$^{-1}$ case, as shown by **??**. Quantitative estimates about maximum vertical winds have to be raised to about 10% and 15% in the 15 m.s$^{-1}$ and 25 m.s$^{-1}$ ambient wind cases respectively, compared to the no-wind estimates (figure not shown).

**6**

**5.1 Sensitivity to subgrid scale diffusion**

Can a difference in the small-scale diffusion schemes in both models be related to the difference of values of PBL depth, maximum TKE and vertical wind speeds detailed in section **??**? In an attempt to answer this question, the LMD LES model has been run with different subgrid scale diffusion coefficients (or mixing coefficients). Tested with much lower values than the typical one used (0.15 at LMD, decreased by a factor up to $10^4$), the PBL diagnostics remain similar in the afternoon (**??**). The same conclusions apply when analyzing the maximum TKE or wind speeds, confirming that the small-scale mixing coefficient has a negligible impact on the development of the convective PBL in the daytime. This exploration is incomplete because the discretization of primitive equations in dynamical core may cause the dynamical core to be naturally diffusive (see section **??** for further details), which cannot be controlled by the above-mentioned subgrid-scale mixing coefficients.

**5.2 Sensitivity to domain size**

The  LES results can be sensitive to domain size, as was described in section **??**. We ran the model with a much wider domain defined by a 250x250x250 grid, and a model top at 16 km instead of 12 km (horizontal and vertical resolutions remain the same than in the reference case). Results show an increase of the PBL depth of 20-25% (**??**), which is not sufficient to explain the 100% differences observed between both LMD and SwRI LES – although it might explain part of the discrepancy.

**6 Challenges of LES intercomparison and suggestions for future studies**

This section aims to identify the key challenges of LES intercomparisons and gives suggestions for a possible path forward for future LES studies.

A first difficulty is to determine which models shall be selected to be intercompared. Ideally, the best intercomparison would include all existing models of the community. However, the more models are included, the more multi-dimensional the intercomparison study becomes, with exponential human, time and computing resources needed. Furthermore, the fact that the models have been built independently may lead to several issues: a specific parameter to be explored in the intercomparison may not exist, or may not be easily accessed in all of them; the models have their own strategy for numerical stability, and imposing a unique setting for all the involved models might be detrimental to the quality of their diagnostics; the models use different options, inputs, and outputs (their initialization might actually be a problem on its own within the intercomparison project); the vertical grid of the models may differ, with heights fixed above ground in a sigma-z formulation much different than the sigma-p coordinate used in other models (possibly leading to spurious effects related to one or the other choices for vertical coordinates). A straightforward solution could be to first compare models sharing the same dynamical core, in order to identify any difference related to the physical packages (radiative transfer, mixing, etc...), and then extend the intercomparison to models built on distinct dynamical cores.

One of the difficulties encountered in this paper is the matching of the near-surface temperatures. Surface layer schemes in respectively the LMD and SwRI LES models are based on distinct formulations; we suspect the impossibility to match near-surface temperatures between the two models with our radiative transfer explorations, which suggests that the impact of those differences in surface layer schemes on the near-surface temperature structure in LES is significant and deserves to be explored by a dedicated study. This impact is difficult to evaluate: it affects the magnitude of turbulent activity near the surface, which in turn modifies the sensible heat flux (through the surface layer schemes), which in turn changes the near-surface temperature profile. Future intercomparison studies should acknowledge this issue and try to converge towards similar near-surface temperature profiles.

Another key difference which has been found between the LMD and SwRI  models is related to subgrid-scale diffusion. This difference of diffusion is suspected to significantly impact the intensity of the convection within the PBL. Consequently, matching (or, if ever possible, deactivation) of subgrid-scale diffusion schemes between models is an important task to realize consistent intercomparisons. This task is more ambitious than it seems. One difficulty is that subgrid-scale numerical diffusion does not depend on only one parameter. For instance, the LMD LES is based on the WRF dynamical core which is inherently diffusive owing to the chosen discretization of the primitive equations (a diffusive term is added for odd-order advection operators (**?**)). Thus, it is not trivial to completely disable subgrid-scale mixing in a model – in this paper, we tested the LMD LES model sensitivity to the subgrid mixing coefficient, but it has little effect on the results since diffusion terms remain inherent to the formulation of the dynamical core (see **??**). Future LES intercomparisons should further investigate the impact of the subgrid-scale mixing on Martian LES results, which appears of central importance in LES and mesoscale models, and probably even in Global Climate Models. This question has remained eluded in most Martian modeling studies to date.

465     Another challenge for Martian LES is the size of the modeling domain. As stated by **?**, the horizontal size of the domain must be large enough so that the periodic boundary conditions cannot influence the turbulence computed for the domain interior. Generally speaking, the length of the grid should be three times the size of the largest eddy that will be resolved by the simulation. Since Martian PBL depth typically reach 10 km, the grid size should be around 30 km. In addition, in order to resolve the smaller eddies, the horizontal resolution must be fine enough, typically around tens of meters. Consequently,

470 for a 30 km square LES domain with a 50 m gridspacing, the number of computational locations would then be 600x600. This is challenging to achieve, although largely within reach of modern supercomputers as is demonstrated by a recent study about statistics of dust devils in Martian LES (**?**). In this paper, the model sensitivity to the size of the domain has been tested and results show an increase of PBL depth about 20-25% (see **??**), in line with previous LMD LES studies ((**?**)). Future intercomparisons should be careful about the domain size and top, and test their model with different configurations.

475     Finally, regarding the strategy we adopted to avoid a comparison of radiative schemes, one suggestion for future Martian LES intercomparisons would be to compare the models with no dust loading at all. This has not been considered in this paper, because the context of the ExoMars mission required LES to be carried out with the most realistic possible temperature profile. LES intercomparisons with a dust-free atmosphere would be a good starting point for future studies – aimed at theoretical discussions and not EDL discussions as in the present paper, since it would enable to compare surface temperatures without

480 the complication of the radiative properties of airborne dust.

**7   Discussion**

The steps of the comparison between the LMD and SwRI LES models can be summarized as follows: LES have been performed at ExoMars landing site and date using settings as similar as possible. A tuning of the LMD radiative transfer routine has been necessary to ensure a similar radiative response of both LMD and SwRI models to same forcings. This tuning involved slight

485 changes of dust properties (extinction and brightness) and surface thermal inertia (increased from 238 tiu to 300 tiu in the LMD LES), essential to obtain similar daytime surface temperatures in both LES and thus to similarly force the turbulence within the PBL.

    The comparison of LES shows similar qualitative results (vertical wind organized into polygonal cells, horizontal gusts) but different quantitative results. SwRI results show values of heat flux, kinetic energy, updraft  and downdraft speeds which

490 are more dispersed, with maximum values higher than LMD results with a ratio between 1.5 and 2, as summarized by Table **??** and Table **??**. This leads to an almost twice more vigorous PBL in the SwRI LES than in the LMD PBL, even though the maximum values only represent a very small fraction of the domain. Results remain similar with or without ambient wind.

    It is important to note that all the results and values obtained from both models remain realistic (with the caveat in mind that no measurements of vertical wind in the Martian convective boundary layer are available from previous missions).

495

    The simulations performed with the LMD LES remain typical compared to previous studies performed with the same model. As an example, the PBL height obtained is in the 5-7 km range of what has been obtained in previous studies in conditions

close to the ExoMars landing site  (e.g similar surface pressure values, see **?**, figure 2 cases b and i). In addition, these values are consistent with radio-occultation measurements (**?**).The boundary layer depths predicted by the SwRI LES model, although clearly in the upper range, are still consistent with those measurements.

The discrepancies observed between both LES  cannot be explained by differences in boundary conditions or radiative forcing, which are similar in both models (less than 1 K difference). Identifying exactly the origin of those discrepancies is challenging (see section **??**) and beyond the scope of the present study, which is only a first step toward an intercomparison of Martian mesoscale and LES models. In this study, we identified the discrepancies between two Martian LES and drew possible future areas of research to disentangle the causes underlying those discrepancies. It has been found in  section **??** that SwRI results exhibit a higher variability at a very small scale than LMD results. This could stem from different assumptions in the subgrid-scale diffusion schemes adopted in both LES. On the one hand, the SwRI LES reflects a much weaker subgrid-scale diffusion than the LMD LES, which would put the SwRI LES at greater risk to overestimate maximum vertical winds ("noisy" turbulent signals). On the other hand, the LMD LES appears to have a subgrid-scale diffusion which efficiently removes the accumulation of energy at the grid point scale, although the possibility still exists that this subgrid-scale diffusion might be too strong, yielding underestimated vertical winds. We tested different subgrid scale diffusion coefficients with the LMD LES model but these changes did not significantly affect the PBL. Consequently, it is plausible that the differences observed between both models lie deeper in the assumptions of their small-scale diffusion schemes, in the inherent diffusion within the dynamical cores (e.g., the advection operators and possibly other numerical diffusion assocated with different schemes) or in other dynamic parts of the models (e.g.  distinct discretizations of the hydrodynamical equations). This remains to be explored further in the respective models. Finally, as suggested in  section **??** and **??**, differences in the PBL near-surface scheme used in both models could also impact the LES results.

 Results in section **??** show that the PBL can be strongly affected by large changes of dust loading and surface conditions. In contrast, although the influence of ambient wind on the Martian PBL turbulence is a significant component to be taken into account for EDL studies, results with ambient wind will not change drastically compared to no-wind simulations (here the Martian situation is quite different from the Earth due to the radiative control of the boundary layer). Comparing windy simulations (**??**, **??**) with extreme soil simulations (**??**, **??**) or with dusty simulations (**??**, **??**) shows that windy conditions represent a secondary influence of Martian PBL convection and  are likely to be overcome by changes in the primary forcing of the Martian PBL, that is, radiative control. Turbulent convective activity is enhanced in windy conditions, but this is overwhelmed by the strong suppression of boundary layer growth caused by very dusty conditions.

The conclusions of the  comparison campaign presented in this paper do not prevent LES from being relevant tools to study the PBL turbulence on Mars and to provide constraints to assess atmospheric hazards encountered by future landing systems, provided caution is exerted along the quantitative lines drawn by the estimates in this paper. Above all, improving the diagnostics provided by Martian LES will require more complete observations of the Martian PBL turbulence in future in situ missions to Mars.

**8 Data availability**

The SwRI and LMD files, figure data or any other source data of this paper are freely available upon request by contacting T.B. or A.S. (tanguy.bertrand@lmd.jussieu.fr, spiga@lmd.jussieu.fr).

535 *Author contributions.* Author contribution

T.B.  and A.C. prepared the comparison of models. T.B. performed the LMD simulations, S.R performed the SwRI simulations. Both T.B. and A.S  wrote the manuscript, with contributions from S.R., E.M. and F.F.

*Competing interests.* Competing interests

540 The authors declare no competing financial interests.

[Figure]

**Figure 5.**  Surface temperatures in SwRI (black) and LMD (red) LES models.

[Figure]

**Figure 6.** LMD (left) and SwRI (right) horizontal section of vertical velocity at about 250 m (top) and at about 2 km (bottom) above the surface, at local time 11:00. Simulations without ambient wind.

[Figure]

**Figure 7.** LMD (left) and SwRI (right) horizontal section of horizontal wind amplitudes at first level above surface (top) and at about 250 m above surface (bottom), at local time 11:00. Simulation without ambient wind

[Figure]

**Figure 8.** LMD (left) and SwRI (right) horizontal section of horizontal wind amplitudes at first level above surface (top) and at about 250 m above surface (bottom), at local time 11:00. Simulation with 15 m.s$^{-1}$ westward ambient wind.

[Figure]

**Figure 9.** Horizontal wind profile in the SwRI (black) and LMD (red) LES models, between local times 11:00-12:00 in the simulation with 15 m.s$^{-1}$ ambient wind.

[Figure]

**Figure 10.** LMD and SwRI variations of boundary layer depth between local times 0908:00 and 17:00 and altitudes above ground 0 and 9 km. Simulations without ambient wind and with 15 m.s$^{-1}$ ambient wind.

[Figure]

**Figure 11.** LMD (left) and SwRI (right) variations of turbulent kinetic energy between local times 07:00 and 19:00 and altitudes above ground 0 and 9 km. Simulations without ambient wind (top) and with 15 m.s$^{-1}$ ambient wind (bottom).

[Figure]

**Figure 12.** LMD (left) and SwRI (right) maximum speeds for convective updrafts reached in the simulation domain between local times 11:00 and 17:00 and altitudes above ground 0 and 8 km. Simulations without ambient wind (top) and with 15 m.s$^{-1}$ ambient wind (bottom).

[Figure]

**Figure 13.** LMD (left) and SwRI (right) maximum speeds for convective downdrafts reached in the simulation domain between local times 11:00 and 17:00 and altitudes above ground 0 and 8 km. Simulations without ambient wind (top) and with 15 m.s$^{-1}$ ambient wind (bottom).

[Figure]

**Figure 14.** Histogram of vertical wind speeds for local times 13:00-14:00 and altitudes 250-5000 m. LMD results are in green, SwRI results are in blue. No ambient wind.

[Figure]

**Figure 15.** Histogram of vertical wind speeds for local times 13:00-14:00 and altitudes 250-5000 m. LMD results are in green, SwRI results are in blue. With 15 m.s$^{-1}$ ambient wind.

[Figure]

**Figure 16.** Evolution of the boundary layer depth for different dust optical depths (LMD LES)

[Figure]

**Figure 17.** Evolution of the maximum turbulent kinetic energy for different dust optical depths (LMD LES)

[Figure]

**Figure 18.** Evolution of the boundary layer depth for different surface albedo values (LMD LES)

[Figure]

**Figure 19.** Evolution of the maximum turbulent kinetic energy for different surface albedo values (LMD LES)

[Figure]

**Figure 20.** Evolution of the maximum turbulent kinetic energy for different large scale ambient winds (LMD LES)

[Figure]

**Figure 21.** Evolution of the boundary layer depth for different large scale ambient winds (LMD LES)

[Figure]

**Figure 22.** Evolution of the  boundary layer depth for different  subgrid scale  mixing coefficient and a case of larger domain (LMD LES). The fist part of the curve of the larger domain simulation is missing because of technical issues but this does not alter the comparison made in the afternoon.